# Visual pursuit behavior in mice maintains the pursued prey on the retinal region with least optic flow

**Carl D Holmgren[1][†], Paul Stahr[1][†], Damian J Wallace[1], Kay-Michael Voit[1], Emily J Matheson[1], Juergen Sawinski[1], Giacomo Bassetto[1,2], Jason ND Kerr[1]\***

[1]Department of Behavior and Brain Organization, Research center caesar, Bonn, Germany; [2]Machine Learning in Science, Eberhard Karls University of Tübingen, Tübingen, Germany

**Abstract** Mice have a large visual field that is constantly stabilized by vestibular ocular reflex (VOR) driven eye rotations that counter head-rotations. While maintaining their extensive visual coverage is advantageous for predator detection, mice also track and capture prey using vision. However, in the freely moving animal quantifying object location in the field of view is challenging. Here, we developed a method to digitally reconstruct and quantify the visual scene of freely moving mice performing a visually based prey capture task. By isolating the visual sense and combining a mouse eye optic model with the head and eye rotations, the detailed reconstruction of the digital environment and retinal features were projected onto the corneal surface for comparison, and updated throughout the behavior. By quantifying the spatial location of objects in the visual scene and their motion throughout the behavior, we show that the prey image consistently falls within a small area of the VOR-stabilized visual field. This functional focus coincides with the region of minimal optic flow within the visual field and consequently area of minimal motion-induced image-blur, as during pursuit mice ran directly toward the prey. The functional focus lies in the upper-temporal part of the retina and coincides with the reported high density-region of Alpha-ON sustained retinal ganglion cells.

**\*For correspondence:**
jason.kerr@caesar.de

[†]These authors contributed equally to this work

**Competing interests:** The authors declare that no competing interests exist.

## Introduction

The visual system of mice serves a variety of seemingly opposing functions that range from detection of predators, to finding shelter and selection of food and mates, and is required to do so in a diverse set of environments (*Boursot et al., 1993*). For example, foraging in open areas where food is available involves object selection, and in the case of insect predation (*Badan, 1986*; *Tann et al., 1991*), involves prey tracking and capture (*Hoy et al., 2016*; *Langley, 1983*; *Langley, 1984*; *Langley, 1988*), but the visual system can also simultaneously be relied on for avoidance of predation, particularly from airborne predators (*Hughes, 1977*). Like with many ground-dwelling rodents (*Johnson and Gadow, 1901*), predator detection in mice is served by a panoramic visual field which is achieved by the lateral placement of the eyes in the head (*Dräger, 1978*; *Hughes, 1979*; *Oommen and Stahl, 2008*) combined with monocular visual fields of around 200 degrees (*Dräger and Olsen, 1980*; *Hughes, 1979*; *Sterratt et al., 2013*). In mice, the panoramic visual field extends to cover regions above the animal's head, below the animal's snout and laterally to cover ipsilaterally from behind the animal's head to the contralateral side, with the overlapping visual fields from both eyes forming a large binocular region overhead and in front of the animal (*Hughes, 1977*; *Sabbah et al., 2017*). In addition, eye movements in freely moving mice constantly stabilize the animal's visual field by counteracting head rotations through the vestibulo-ocular reflex (VOR) (*Meyer et al., 2020*; *Meyer et al., 2018*; *Michaiel et al., 2020*; *Payne and Raymond, 2017*)

**eLife digest** Mice have a lot to keep an eye on.

To survive, they need to dodge predators looming on land and from the skies, while also hunting down the small insects that are part of their diet. To do this, they are helped by their large panoramic field of vision, which stretches from behind and over their heads to below their snouts.

To stabilize their gaze when they are on the prowl, mice reflexively move their eyes to counter the movement of their head: in fact, they are unable to move their eyes independently. This raises the question: what part of their large visual field of view do these rodents use when tracking a prey, and to what advantage?

This is difficult to investigate, since it requires simultaneously measuring the eye and head movements of mice as they chase and capture insects. In response, Holmgren, Stahr et al. developed a new technique to record the precise eye positions, head rotations and prey location of mice hunting crickets in surroundings that were fully digitized at high resolution. Combining this information allowed the team to mathematically recreate what mice would see as they chased the insects, and to assess what part of their large visual field they were using.

This revealed that, once a cricket had entered any part of the mice's large field of view, the rodents shifted their head – but not their eyes – to bring the prey into both eye views, and then ran directly at it. If the insect escaped, the mice repeated that behavior. During the pursuit, the cricket's position was mainly held in a small area of the mouse's view that corresponds to a specialized region in the eye which is thought to help track objects. This region also allowed the least motion-induced image blur when the animals were running forward.

The approach developed by Holmgren, Stahr et al. gives a direct insight into what animals see when they hunt, and how this constantly changing view ties to what happens in the eyes. This method could be applied to other species, ushering in a new wave of tools to explore what freely moving animals see, and the relationship between behaviour and neural circuitry.

maintaining the large panoramic overhead view (*Wallace et al., 2013*) critical for predator detection (*Yilmaz and Meister, 2013*).

Given the VOR stabilized panoramic field of view, it is not clear what part of the visual field mice use to detect and track prey (but see: *Johnson et al., 2021*). Mouse retina contains retinal ganglion cells (RGCs), the output cells of the retina, with a broad diversity of functional classes (*Baden et al., 2016*; *Bleckert et al., 2014*; *Franke et al., 2017*; *Zhang et al., 2012*). Given the lateral eye position, the highest overall density faces laterally (*Dräger and Olsen, 1981*; *Sabbah et al., 2017*; *Salinas-Navarro et al., 2009*; *Stabio et al., 2018*). Further, as the functionally defined ganglion cells (*Baden et al., 2016*; *Bleckert et al., 2014*; *Franke et al., 2017*; *Zhang et al., 2012*) and cone subtypes (*Szél et al., 1992*) are segregated into retinal subregions within the large stabilized field of view, recent studies suggest that retinal subregions are tuned for specific behavioral tasks depending on what part of the world they subtend (*Baden et al., 2016*; *Bleckert et al., 2014*; *Hughes, 1977*; *Sabbah et al., 2017*; *Szatko et al., 2020*; *Zhang et al., 2012*).

The challenge is to measure what part of the visual field the mouse is attending to during a visually based tracking task (*Hoy et al., 2016*) and the location of all objects within the field of view during the behavior. While recent studies have implied the relationship between prey and retina through tracking head position (*Johnson et al., 2021*) or measured both the horizontal and vertical eye rotations (*Meyer et al., 2020*; *Meyer et al., 2018*) during pursuit behavior (*Michaiel et al., 2020*) to uncover a large proportion of stabilizing eye-rotations, what is missing is the extent and location of the area used when detecting and pursuing prey, and the relationship to the retina (*Bleckert et al., 2014*).

Here, we measured the position of a cricket in the visual fields of freely moving mice performing a prey pursuit behavior, using head and eye tracking in all three rotational axes, namely horizontal, vertical, and torsional. Eye tracking included an anatomical calibration to accurately account for the anatomical positions of both eyes. To quantify object location in the animal's field of view and generate optic flow fields, head and eye rotations were combined with a high-resolution digital reconstruction of the arena to form a detailed visual map from the animal's eye perspective. Given that

mice use multisensory strategies during prey pursuit (*Gire et al., 2016*; *Langley, 1983*; *Langley, 1988*) and can track prey using auditory, visual, or olfactory cues (*Langley, 1983*; *Langley, 1988*), we developed a behavioral arena that isolated the visual aspect of the behavior by removing auditory and olfactory directional cues to ensure that the behavior was visually guided. To transfer the retinal topography onto the corneal surface, we developed an eye model capturing the optical properties of the mouse eye. We show that during prey detection mice preferentially position prey objects in stable foci located in the binocular field and undertake direct pursuit. Prey objects remain in the functional foci through the stabilizing action of the VOR, and not through active prey-pursuit eye movements. The stabilized functional foci are spatially distinct from the regions of highest total retinal ganglion cell density, which are directed laterally, but coincides with the regions of the visual field where there is minimal optic flow and therefore minimal motion-induced image disturbance during the behavior, as the mouse runs towards the cricket. Lastly, by building an optical model that allows corneal spatial locations to be projected onto the retina, we suggest that the functional foci correspond to retinal subregions containing a large density of Alpha-ON sustained RGCs that have center-surround receptive fields and project to both superior colliculus and dLGN (*Huberman et al., 2008*) and possess properties consistent with the requirements for tracking small and mobile targets (*Krieger et al., 2017*).

## Results

### Forming a view from the animal's point of view

To measure what part of the visual field mice use during prey capture while also considering that mice can use multisensory strategies during prey pursuit (*Gire et al., 2016*; *Langley, 1983*; *Langley, 1988*), we first developed an arena which isolated the visual component of prey pursuit by masking olfactory and auditory spatial cues (*Figure 1A*, see Materials and methods for details). By removing both olfactory and auditory cues, the average time to capture a cricket approximately doubled compared to removal of auditory cues alone (time to capture, median ± SD, control 24.92 ± 16.77 s, olfactory and auditory cues removed, 43.51 ± 27.82 s, p = 0.0471, Wilcoxon rank sum test, N=13 control and 12 cue removed trials from N = 5 mice). To track mouse head and eye rotations during prey capture, we further developed a lightweight version of our head mounted oculo-videography and camera-based pose and position tracking system (*Wallace et al., 2013*; *Figure 1B* and Materials and methods). This approach allowed quantification of head rotations in all three axes of rotation (pitch, roll, and yaw), as well as eye rotations in all three ocular rotation axes (torsion, horizontal, and vertical, *Figure 1C*, *Figure 1—figure supplement 1A and B*). The same camera-based system was used to track and triangulate the position of the cricket (see Materials and methods and *Figure 1—figure supplement 1C*). To quantify the position and motion of the environment and cricket in the mouse's field of view, we also developed a method that enabled a calibrated environment digitization to be projected onto the corneal surface. This approach utilized a combination of laser scanning and photogrammetry, giving a resolution for the reconstruction of the entire experimental room of 2 mm, as well as a detailed measurement of eye and head rotations (*Figure 1D–E*, and see Materials and methods). Mice, like rats (*Wallace et al., 2013*), have a large visual field of view which extends to also cover the region over the animal's head (*Figure 1F*). To ensure the entire visual fields of the mouse could be captured during behavior, we digitized the entire experimental room and contents (*Figure 1E*, *Figure 1—figure supplement 1D–F*, *Video 1*). The coordinate systems of the environmental digitization and mouse and cricket tracking systems were registered using 16–20 fiducial markers identified in both the overhead camera images and the digitized environment. The average differences in position of fiducial points between the two coordinate systems were less than 1 mm (mean ± SD, x position, 0.18 ± 3.1 mm, y position, 0.07 ± 1.6 mm, z position, 0.66 ± 1.8 mm, N=54 fiducial points from three datasets). The next step was to re-create the view for each eye. First, and for each mouse, the positions of both eyes and nostrils were measured with respect to both the head-rotation tracking LEDs and head-mounted cameras, then calibrated into a common coordinate system (*Figure 1B*). Together, this enabled a rendered representation of the digitized field of view for each combination of head and eye rotations. This rendered image, from the animal's point of view, contained all the arena and lab objects (*Figure 1G–H*, *Video 2* and *3*, *Figure 1—figure supplement 1G*). In addition to object position and distance (*Figure 1I*), the motion

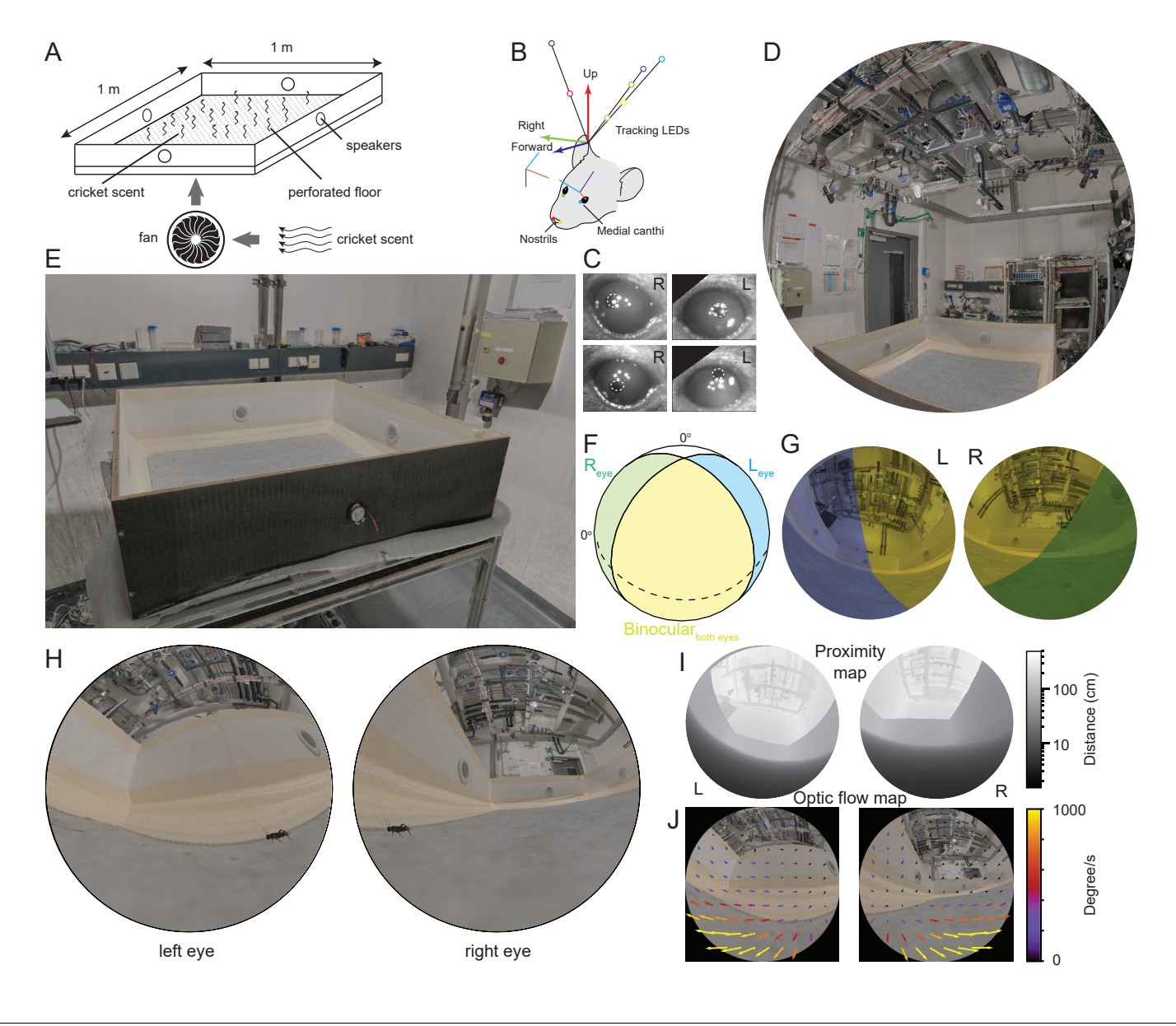

**Figure 1.** Reconstruction of experimental arena and surrounds from the animal's perspective. (**A**) Schematic of experimental arena with olfactory and auditory noise. (**B**) Schematic of tracking, anatomical and eye camera calibration. Head position and orientation was tracked using seven IR-LEDs (colored circles). Nostrils (red, yellow filled circles), left (blue filled circle), and right (green filled circle) medial canthi were identified and triangulated in calibration images and used to define a common coordinate system (forward, blue arrow, right, green arrow, and up, red arrow), into which the calibrated eye camera location and orientation could also be placed (eye camera vertical, cyan, horizontal, purple, camera optical axis, red). (**C**) Example left- and right eye camera images with tracked pupil position (white dashed outlines). (**D**) Rendered digital reconstruction of the laboratory room and (**E**) experimental arena. (**F**) Schematic representation of mouse's left- (blue) and right (green) visual fields, showing also the region of binocular overlap (yellow) and un-seen region (white). (**G**) Reconstruction of the arena and room from the animal's left- and right eye perspective, with monocular and binocular regions colored as in (**F**). (**H**) Reconstruction of the animal's view of the prey (cricket - black) in the experiment arena. (**I**) Representation of left and right eye views of the arena and surrounding objects grayscale-coded by distance from the eye. (**J**) Rendered animal's eye views from the left- and right eyes with overlay of arrows representing optic flow during 10 ms of free motion.

The online version of this article includes the following source data and figure supplement(s) for figure 1:

**Source data 1.** Related to *Figure 1D*.
**Source data 2.** Related to *Figure 1G*.
**Source data 3.** Related to *Figure 1H*.
**Source data 4.** Related to *Figure 1I*.

*Figure 1 continued on next page*

*Figure 1 continued*

**Source data 5.** Related to *Figure 1J*.

**Figure supplement 1.** Generation of mouse eye views during cricket pursuit.

of the environment and each object in the field of view could be quantified as the mouse performed prey capture behaviors (*Figure 1J*, and *Figure 1—figure supplement 1H*).

## During pursuit the image of the prey consistently falls in a localized visual region

Crickets (*Acheta domesticus*), shown previously to be readily pursued and preyed upon by laboratory mice (*Hoy et al., 2016*), provided a prey target that could successfully evade capture for extended periods of time (total time for each cricket before capture: 64.4 ± 39.3 s, average time ± SD, N = 21 crickets and three mice, *Video 4* and *5*). To ensure that only data where the mouse was actively engaged in the detection and tracking of the cricket were used, we identified occasions where the mouse either captured the cricket, or contacted the cricket but the cricket escaped (see Materials and methods for definitions), and then quantified the trajectories of both mouse and cricket leading up to the capture or capture-escape (*Figure 2A*). Within these chase sequences we defined three behavioral epochs (detect, track, and capture, *Figure 2B*, see Materials and methods for definition details) based on the behavior of mouse and cricket, and similar to previous studies (*Hoy et al., 2016*).

Upon cricket detection, mice oriented and ran towards the cricket, resulting in a significant adjustment to their trajectory ($\Delta$ target bearing: 40.2 ± 35.1°, P = 6.20 x 10$^{-10}$, $\Delta$ speed: 10.2 ± 7.4 cm/s, P = 1.91 x 10$^{-10}$; N=57 detect-track sequences N = 3 mice; Paired Wilcoxon's signed rank test for both tests), and a rapid reduction in the Euclidean distance to the cricket (*Figure 2C*). During tracking, the cricket was kept in front of the mouse, resulting in a significant reduction in the spread of target bearings compared to during detect epochs (*Figure 2D*, Target bearing: detect 6.2 ± 62.1°, track: 2.5 ± 25.6°, mean ± SD, Brown-Forsythe test p = 0, *F* statistic=7.05x10$^3$, N=4406 detect and 13624 track frames, N=3 mice), consistent with previous findings (*Hoy et al., 2016*). To avoid the closing phase of the pursuit being associated with whisker strikes (*Shang et al., 2019*; *Zhao et al., 2019*), tracking periods were only analyzed when the mouse was more than 3 cm from the cricket, based on whisker length (*Ibrahim and Wright, 1975*).

Using the detailed digitization of the behavioral arena and surrounding laboratory (*Figure 1E*, *Video 1*), an image of the cricket and objects in the environment was calculated for each head and eye position during the predator-prey interaction (*Video 2 and 3*). Using this approach, we addressed the question of what area of the visual field was the cricket located in during the various behavioral epochs. In the example pursuit sequence in *Figure 2E*, the cricket was initially located in the peripheral visual field and then transitioned to the lower nasal binocular quadrant of the cornea-view during pursuit and capture (red trace in left eye to blue trace in both eyes). Correspondingly, an average probability density map calculated for all animals during the detect epoch showed a

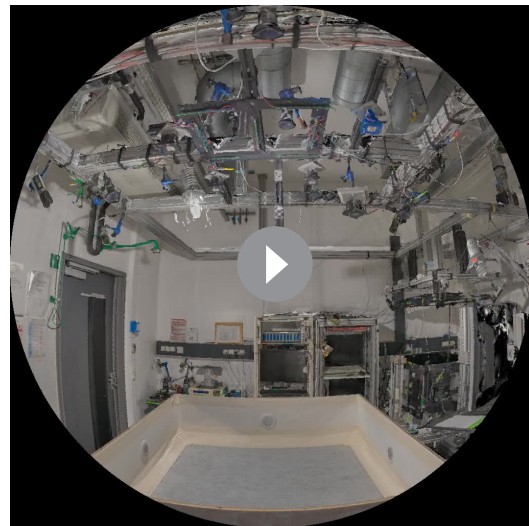

**Video 1.** Digitized and rendered view of the experiment arena and surrounding environment. Laser scanned and digitally reconstructed experiment environmental, providing positional information of objects within the mouse's environment. When combined with the tracked 3D cricket positions and the tracked mouse head and eye positions and rotations this allowed the generation of a frame-by-frame mouse eye view of the prey and the surroundings.

https://elifesciences.org/articles/70838#video1

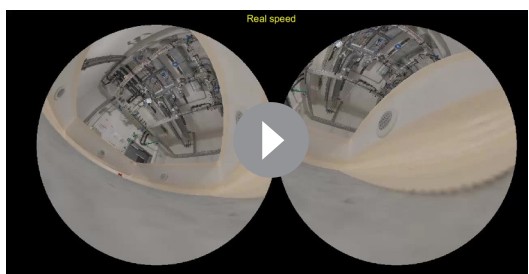

**Video 2.** Reconstruction of the mouse's left and right eye field of view during one example behavioral sequence. Real speed.
https://elifesciences.org/articles/70838#video2

**Video 3.** Reconstruction of the mouse's left and right eye field of view during one example behavioral sequence, as shown in *Video 2*, but slowed to 0.5x real speed.
https://elifesciences.org/articles/70838#video3

very broad distribution of cricket positions across the visual field (*Figure 2F*, *Figure 2—figure supplement 1A and B*). Upon detection the mouse oriented toward the cricket, bringing it toward the lower nasal binocular visual field (*Figure 2E*, *Video 6*). When averaged for all pursuit sequences from all animals, projected cricket positions formed a dense cluster on the cornea of both eyes (*Figure 2G and H*, *Figure 2—figure supplements 1A,C–D*, 50% contour center for left and right eye respectively, radial displacement from optical axis 64.3 ± 7.5° and 63.3 ± 9.9°, rotational angle 126.2 ± 8.9° and −115.7 ± 6.1°, mean ± SD, N = 3 mice), which was significantly different from the cluster in the detect epoch (average histogram of the location of cricket image during tracking phase vs average histogram of the location of cricket during detect phase: Left eye P = 3.54 x $10^{-46}$, Right eye P = 1.08 x $10^{-81}$, differences calculated by taking the Mean Absolute Difference with bootstrapping, N=57 detect-track sequences, N = 3 mice). Thus, during the tracking and pursuit behavior the image of the prey consistently fell on a local and specific retinal area that we refer to from here on as the functional focus. The functional focus fell within the binocular field, while the region of elevated density of RGCs has been found to be located near the optical axis (*Dräger and Olsen, 1981*), which suggests that the location of the retinal specialization may not overlap with the functional focus.

## Relative locations of functional foci and ganglion cell density distributions

To establish the relation between the identified functional focus and the density distribution of RGCs, we made a mouse eye-model (*Figure 3A*), modified from previous models (*Barathi et al., 2008*). Using the eye model, retinal spatial locations could be projected through the optics of the mouse eye to the corneal surface. We first reconstructed the isodensity contours quantifying the distribution of all RGCs (*Dräger and Olsen, 1981*) to define the retinal location with the highest overall ganglion cell density (*Figure 3—figure supplement 1A–C*, note that these contours are also in agreement with other recently published maps of total RGC density [*Bleckert et al., 2014*; *Zhang et al., 2012*]). The lens optical properties were based on a GRIN lens (present in both rats [*Hughes, 1979*; *Philipson, 1969*] and mice [*Chakraborty et al., 2014*]). To determine the optical characteristics of this lens, we developed a method which combined models of the lens surface and refractive index gradient (*Figure 3A*,

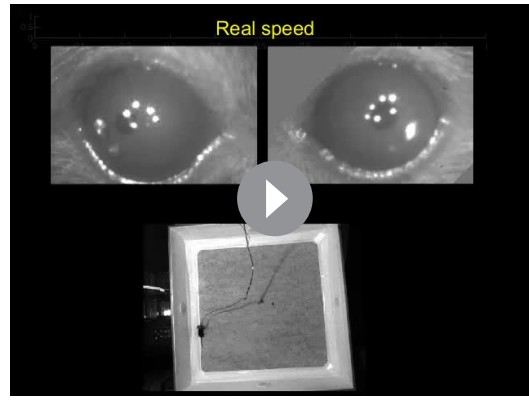

**Video 4.** Left and right eye camera images and one overhead camera view showing one complete cricket pursuit, from shortly after release of the cricket into the arena to cricket capture. Real Speed.
https://elifesciences.org/articles/70838#video4

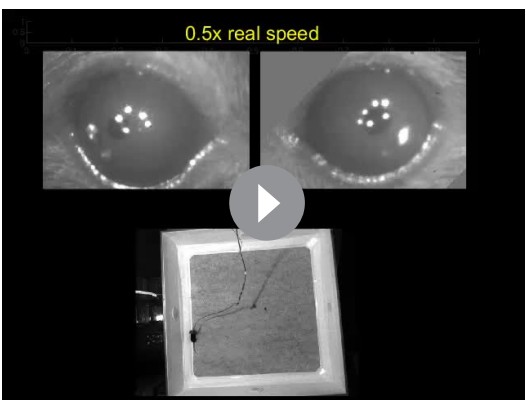

**Video 5.** The same cricket pursuit as shown in *Video 4* but slowed to 0.5x real speed.
https://elifesciences.org/articles/70838#video5

*Figure 3—figure supplement 1D* and *Tables 1* and *2*, see Materials and methods for details). Using this model, the contours representing the retinal specializations were projected through the eye model onto the corneal surface to determine equivalent corneal locations (*Figure 3B*, *Figure 3—figure supplement 1E*). Comparing this location to the functional focus location showed that the region with the highest overall RGC counts and the functional focus (*Figure 3B*) occupied distinct retinal locations (*Figure 3C*). Viewed from above the animal's head, the functional foci were directed at the region in front of the animal's nose and within the region of stable binocular overlap (azimuth: 1.4 ± 8.8° and −4.4 ± 9.3°, elevation 5.7 ± 2.1° and 4.9 ± 1.4° for left and right eyes respectively, N = 13641 frames, N=3 mice), while the retinal specialization was directed laterally (azimuth: −66.2 ± 6.7° and 70.3 ± 4.7°, elevation: 30.8 ± 12.2° and 41.0 ± 13.5° for left and right eyes respectively, N = 13641 frames N=3 mice. *Figure 3D*, *Figure 3—figure supplement 1F–G*). Given that density distributions for different subtypes of RGCs can be spatially heterogeneous with density peaks in distinctly different retinal locations, and that the region of peak density for Alpha-ON sustained RGC's is spatially located on the dorso-temporal retina (*Bleckert et al., 2014*), consistent with projecting to the front of the animal, we next quantified whether this region overlapped with the functional focus observed here (*Figure 3E*).

The average 50% contour of the functional focus was overlapped by the highest density of Alpha-ON sustained RGC's by 35% and 67% for left and right eye respectively (*Figure 3E*, black, mean ± SD for left and right eye, 35.1 ± 19.8%, 66.7 ± 0.09%, p = 0.095 and 0.019, one-sided Student's t-test), and the overlap with the second highest density was 83% and 95% (mean ± SD for left and right eye, 82.8 ± 20.1%, 94.8 ± 24.7%, p = 0.042 and 0.003, one-sided Student's t-test), suggesting a high degree of correspondence between the highest density of Alpha-ON sustained RGC's and the functional focus during pursuit behavior. Viewed from above the animal's head the functional foci were directed at the region in front of the animal's nose azimuth: 1.4 ± 8.8° and −4.4 ± 9.3°, elevation: 5.7 ± 2.1° and 4.9 ± 1.4° for left and right eyes respectively, N = 13641 frames, N=3 mice. The Alpha-ON sustained RGC's were also directed in front of the animal's nose (mean ± SD, elevation:16.0 ± 6.9° and 10.8 ± 11.0°, azimuth: −3.6 ± 0.7° and 5.8 ± 7.9° for left and right eyes respectively, N = 168400 frames, N = 3 mice, *Figure 3F*). Together this suggests that objects falling within the functional foci are processed by Alpha-ON sustained RGC's.

## Combination of torsional, horizontal, and vertical eye rotations counter head rotations

Eye movements in freely moving mice, like with rats (*Wallace et al., 2013*), can be large and rapid (*Meyer et al., 2020*; *Payne and Raymond, 2017*), and counter head rotations through the VOR, enabling the large field of view around the animals head to be stabilized while the animal is moving. The relationships between head rotations and both the horizontal and vertical eye rotations have recently been quantified, and in addition it has been reported that both during exploration and hunting, mice also have abrupt gaze shifts brought about by the combination of head rotation and conjugate saccade-like horizontal eye rotations (*Meyer et al., 2020*; *Michaiel et al., 2020*). We also observed both forms of eye movements in the current study (*Figure 4—figure supplement 1*). However, how these rotations combine with torsional rotations is not known. If mouse VOR operates similar to that observed in the rat (*Wallace et al., 2013*), torsional rotations in the mouse will play a significant role in stabilizing the visual field particularly during changes in head pitch. As with the vertical and horizontal rotations (*Meyer et al., 2018*), torsional rotations in freely moving mice spanned a wide range of rotation angles (*Figure 4—figure supplement 2A–D*), and were correlated with head pitch (Pearson's correlation coefficient (r): detect −0.72, 0.58, track: −0.60 and 0.53 for left

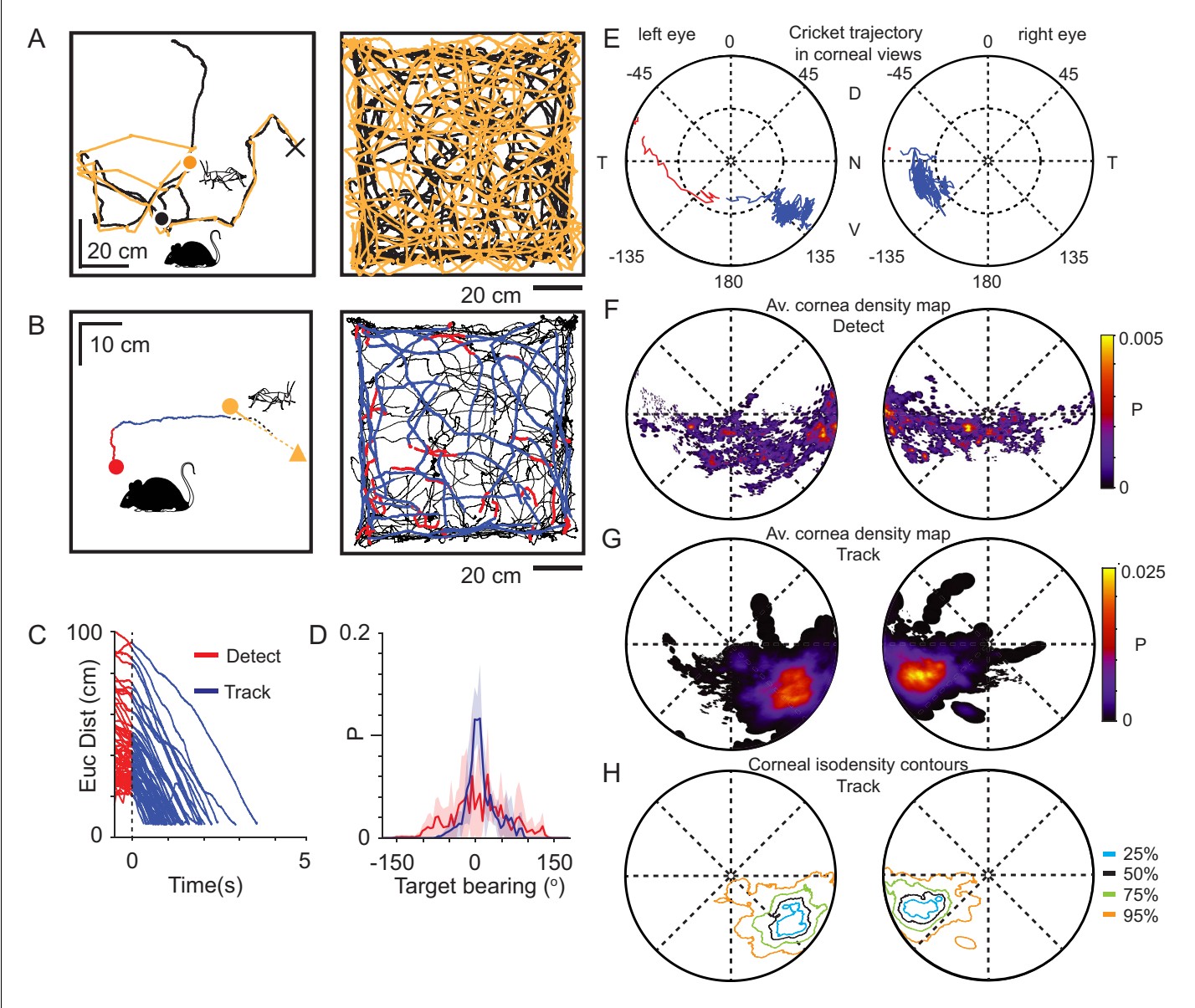

**Figure 2.** Mice use a focal region of their visual field to track prey. (**A**) Mouse (black) and cricket (orange) paths during a single pursuit sequence (left), and for all pursuit sequences in one session for one animal (right). Pursuit start denoted as filled circles and cricket capture as X. (**B**) Mouse (red and blue) and cricket (orange) paths during an individual pursuit sequence (left) and all pursuit sequences in one session from one animal (right), showing detect (red) and track (blue) epochs of the mouse path. Paths after a cricket escape shown dashed. Pursuit sequence start shown as filled circles, cricket landing point after a jump shown as a filled triangle. (**C**) Euclidean distance between mouse and cricket during detect (red) and track (blue) epochs (n=65 trajectories, n=3 mice). (**D**) Mean and SD bearing to cricket (angle between mouse's forward direction and cricket location) during detect (red), and track (blue) epochs from all animals (detect: 57 epochs; track: 65 epochs, n=3 animals, bin size = 5°). (**E**) Trajectory of the projected cricket position in the left and right corneal views, during a single pursuit sequence. Color scheme as for D. The inner dashed circle is 45° from the optical axes. Dorsal (D), ventral (V), nasal (N), and temporal (T) directions indicated. (**F**) Average probability density maps for detect epochs (4628 frames from three animals). Orientation as in E. (**G**) Average probability density maps for track epochs (13641 frames from three animals). Orientation as in E. (**H**) Isodensity contours calculated from the average probability density maps for track epochs. (note that 50% means that this region contains 50% of the total density, and likewise for the other contours). Orientation as in E.

The online version of this article includes the following source data and figure supplement(s) for figure 2:

**Source data 1.** Related to *Figure 2A,B,C,D,H*.

**Source data 2.** Related to *Figure 2E*.

**Source data 3.** Related to *Figure 2F*.

**Source data 4.** Related to *Figure 2G*.

*Figure 2 continued on next page*

*Figure 2 continued*

**Figure supplement 1.** Individual corneal prey image heatmaps.

and right eyes respectively, N=4406 detect and 13624 track frames, N=3 mice, *Figure 4—figure supplement 2C–D*) as well as head roll (Pearson's correlation coefficient (r): detect: −0.46, −0.47 track: −0.45 and −0.48 for left and right eyes respectively, N=4406 detect and 13624 track frames, N=3 mice, *Figure 4—figure supplement 2L–M*), as found previously for freely moving rats (*Wallace et al., 2013*). As with rats, the rotational relationship between the two eyes was dynamic with different forms of coordination (*Figure 4—figure supplement 2E–I*), including episodes of in- and excyclovergence (torsional rotation of both eyes toward or away from the nose, respectively) as well as dextro- and levocycloversion (torsional rotation of both eyes to the animal's right or left, respectively). We next analyzed how effectively rotations of the eye around multiple rotational axes combined to compensate the rotation of the head (*Figure 4A*, *Figure 4—figure supplement 3A–G*). We compared movement of the head around its rotational axes and eye movements around the same rotational axes (*Figure 4A*), effectively defining alternative rotational axes for the eyes to match the axes of the head. Rotation of the eye around these re-defined axes would involve simultaneous rotations in multiple of the eye's anatomical axes. The gain of this compensation was relatively linear and less than unity for both pitch- and roll-axes, indicating on average under-compensation of the head rotation (slope (gain) of relation for pitch axis, −0.45 ± 0.12 and −0.48 ± 0.06; roll axis −0.51 ± 0.12 and −0.62 ± 0.05 for left and right eye respectively, 168852 frames, N=3 mice). The relatively linear relationships between head and eye rotation around the head pitch and roll axes (*Figure 4B*) with a transition through the origin suggests that the horizontal, vertical and torsional eye movements are combined to effectively compensate pitch- and roll-related head movements. We next digitally froze each individual eye rotation axis (torsion, vertical, and horizontal) and measured the effect on countering the head rotation (*Figure 4C*). For rotations around the head x-axis (head pitch changes) the gain of compensation was most affected by freezing torsional rotations (*Figure 4C*, gain mean ± SD, control: −0.45 ± 0.12and-0.48 ± 0.06; torsion frozen −0.24 ± 0.1 and −0.24 ± 0.01, for left and right eyes respectively, N = 168852 frames, N=3 mice), while freezing vertical or horizontal rotations had more minor effects (*Figure 4C*, *Table 3*). The gain of compensation for rotations around the head y-axis (head roll changes) was dramatically affected by freezing vertical rotations (*Figure 4C*, gain mean ± SD, control: −0.51 ± 0.12 and −0.62 ± 0.05, vertical frozen −0.16 ± 0.14 and −0.17 ± 0.03, for left and right eyes respectively, N = 168852 frames, N=3 mice), with freezing torsion also reducing compensation gain but to a lesser extent (*Figure 4C*, *Table 3*). We next quantified the stability and alignment of the animal's binocular visual field during the pursuit sequences and determined the location of the functional foci within the stabilized region.

## Functional foci are located in the motion-stabilized binocular visual field

Similar to rats, left and right visual fields overlapped extensively (*Dräger and Olsen, 1980*; *Hughes, 1979*), with eye movements creating variability in the extent of the overlap at the edges of the two visual fields, the transition from monocular to binocular (*Figure 4D*). The functional foci for both eyes were predominately contained within the region of continuous binocular overlap. A horizontal transect through the optical axis for all animals showed a gradual transition from continuous binocular coverage to zero binocular coverage commencing just nasal of the optical axis (*Figure 4D*, *Figure 4—figure supplement 3H and I*), indicating that the region of highest overall RGC density spans this transitional region whereas the functional foci are, on

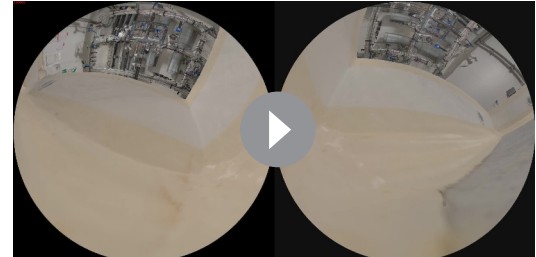

**Video 6.** Reconstructed left and right mouse-eye views for one example pursuit behavioral sequence, showing the trajectory of the cricket position in the eye views during the detect (red) and track (blue) segments of the behavior.

https://elifesciences.org/articles/70838#video6

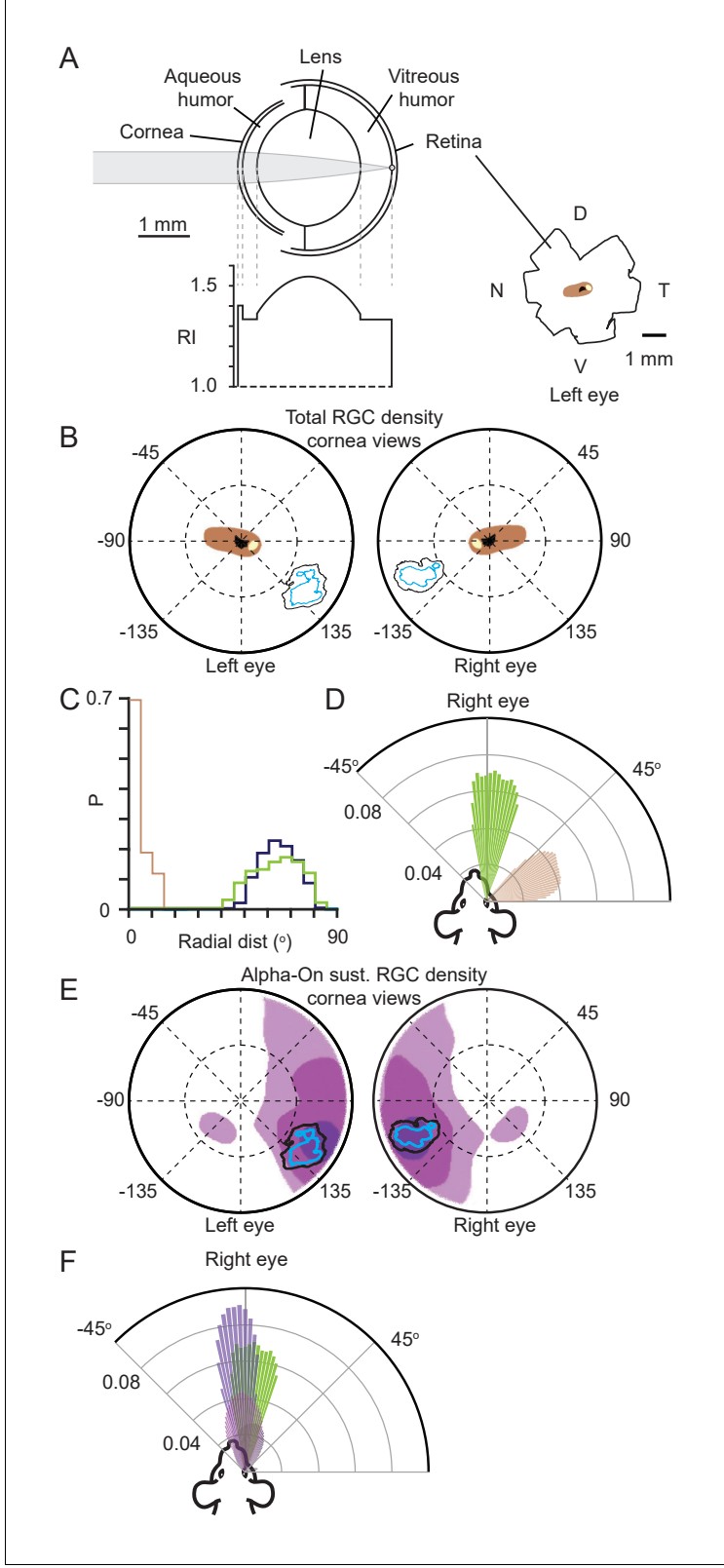

**Figure 3.** Functional foci are not sampled by the highest density retinal ganglion cell region. (A) Schematic of mouse eye model (left upper) with profile of all refractive indices (RI, left lower). Reconstructions of the optic disc (black), highest (>8000 cells/mm², beige) and second highest (>7000 cells/mm², brown) retinal ganglion cell (RGC) density regions redrawn from *Dräger and Olsen, 1981*, shown in lower right. (B) Position in corneal views of the

*Figure 3 continued*

high RGC density regions (brown and beige filled regions), and isodensity contours from *Figure 2H* after projection through the eye model. Orientation as in *Figure 2E*. (C) Horizontal axis histograms for the nasal half of the corneal view of the second highest RGC region (brown) and 50% isodensity contour for left (blue) and right (green) eyes. (D) Top-down view of the coverage regions for the right eye of the 50% isodensity contour (green, N = 7551 frames) and second highest RGC region (brown, N = 51007 frames) for a single animal. Bars represent the probability density function for the respective regions at that azimuth angle. (E) Position in corneal views of Alpha-ON sustained RGC densities (redrawn from *Bleckert et al., 2014*) after projection through the eye model. Colored regions show the 95% (dark purple), 75% (medium purple), and 50% (light purple) contour regions of the peak Alpha-ON sustained RGC density. Isodensity contours from *Figure 2H*. (F) Top-down view of the coverage regions for the right eye of the 95% (dark purple), 75% (medium purple), and 50% (light purple) Alpha-ON sustained RGC contour regions (same as in E, N = 51007 frames) and the 50% isodensity contour from D (green) for a single animal. For the Alpha-ON sustained RGC contour regions, 50% means that this region contains all points which are at least 50% of the peak RGC density.

The online version of this article includes the following source data and figure supplement(s) for figure 3:

**Source data 1.** Related to *Figure 3A*.
**Source data 2.** Related to *Figure 3B*.
**Source data 3.** Related to *Figure 3C*.
**Source data 4.** Related to *Figure 3D*.
**Figure supplement 1.** Projecting high retinal ganglion cell density region from retina to cornea.

---

average, contained within the binocular region (*Figure 4—figure supplement 3H*).

We next quantified the variability of alignment of the left and right visual fields within the binocular region, and specifically in the functional focus location (*Figure 4E*) by using the center of mass (50% isodensity contour center) of the left eye functional focus as an initial reference point and projecting this point to the boundary of a hypothetical sphere surrounding the head. This contact point on the sphere was then re-projected into the right eye to identify the matching location of the left eye (*Figure 4E*). We then followed the trajectory of the re-projected point in the right eye to get a measurement of alignment variability (*Figure 4F*, for comparison with the locations in the right eye projected into the left eye see *Figure 4—figure supplement 3I–K*). While pursuing crickets, alignment precision varied through time (*Figure 4G*) with the mean alignment of the reference point over all animals and data segments being ~8–9°, which is around the size of V1 cortical neuron receptive fields (~5–15° [*Niell and Stryker, 2008*], *Figure 4H*, mean ± SD, left eye projected into right eye 8.8 ± 6.9°; right eye projected into left eye 8.6 ± 6.7°). Repeating this analysis for all points within the region where the probability of binocular overlap was greater than 5% showed that there was a relatively uniform alignment over the entire region (*Figure 4I*), and that the average alignment error in the functional foci was 8–10°. Coordination of eye movements was important for alignment, as freezing the movements of one eye to its mean position resulted in a significant increase in the alignment error when comparing individual cricket tracking sequences (left all rotations vs. left eye frozen P = 1.78 x $10^{-10}$, right eye all rotations vs. right eye frozen P = 7.12 x $10^{-11}$, N=52 sequences, unpaired Student's t-test), and a ~54% increase in the mean alignment error over all frames for the reference location (*Figure 4I*, left eye projected into right eye (left eye frozen) 13.4 ± 8.3°; Right eye

---

**Table 1.** Mouse eye model curvatures.
Radii of curvature of the optical components of the mouse eye model in *Figure 3A*.

| Ocular Component | Radius of curvature ($\mu$m) |
| --- | --- |
| Anterior Cornea | −1408* |
| Posterior Cornea | −1372* |
| Anterior Lens | 1150* |
| Posterior Lens | 1134* |
| Retina | 1598* |

\* Values from *Barathi et al., 2008*.

**Table 2.** Mouse eye model thicknesses and refractive indices.

| Ocular Component | Thickness($\mu m$) | Index of refraction |
|---|---|---|
| Cornea | 92* | 1.402* |
| Anterior chamber | 278* | 1.334* |
| Lens | 2004* | 1.36–1.55[†] |
| Vitreous chamber | 609* | 1.333* |

\* Values from **Barathi et al., 2008**.

† Minimum and maximum values after eye model optimization.

projected into left eye (right eye frozen) 13.4 ± 8.3°, mean ± SD, 159318 frames, N=3 mice), which also resulted in a uniform increase in alignment error over the whole overlap region (*Figure 4J* and *Figure 4—figure supplement 3J–L*). In summary, during pursuit behavior the functional foci are located in a stable binocular region of the mouse's visual field. However, in the absence of a mechanism for voluntarily directing its gaze toward a specific target, such as smooth pursuit, tight coupling of VOR-evoked eye movements to head rotations would seem to be restrictive to an animal's ability to move the target into a specific part of their visual field during pursuit. We therefore next measured what mechanisms mice use to bring the prey into their functional focus.

## Behavioral mechanisms for maintaining prey within functional foci

At detection, mice orient toward their target, aligning their head with the prey and running towards it (*Figure 2D*), keeping the cricket within a narrow window around its forward direction. This provides a direct way for mice to hold their prey within their binocular visual fields (*Figure 4D*). We next measured whether additional head or eye movements are used to keep the target within the functional foci. If the mice were actively maintaining the prey within a fixed location of their visual fields, the position of the cricket image should not change as the mouse approaches the cricket. The cricket image location could be maintained by either a head or eye rotation. If they were not actively maintaining the prey in a fixed location, the cricket images should shift downwards in the visual fields as the mouse approaches. To distinguish between these two possibilities, we plotted the cricket positions in the eye views color-coded by the distance between the mouse and cricket (*Figure 5A*). As the mouse approached the cricket during the track behavioral epoch, the projected cricket positions systematically shifted lower in the visual field (*Figure 5A* lower). This suggests that the mice did not use additional head or eye movements (*Figure 5—figure supplement 1*) to bring the cricket into the functional foci, but rather manipulated the cricket's position in the eye view by orienting and moving towards the target. Consistent with this, head pitch remained stable as the mice approached the crickets (*Figure 5B*). Furthermore, there was no significant difference in head pitch as a function of distance to the cricket between non-tracking and tracking periods (non-tracking head pitch: −3.7 ± 26.5°, mean ± SD, median = −11.3°, tracking head pitch: −12.9 ± 15.7°, mean ± SD, median = −14.6°, Ks test, P = 0.709, paired Student's t-test P = 0.197, N = 18 non-tracking and track sequences, N = 3 mice). In addition, and consistent with previous findings (*Michaiel et al., 2020*), mice did not significantly converge their eyes as they approached the crickets (non-tracking head vergence: 7.6 ± 13.5°, mean ± SD, median = 8.6°, tracking head vergence: 2.5 ± 16.7°, mean ± SD, median = 3.2°. Ks test, P = 0.425, paired Student's t-test P = 0.225, N=18 non-tracking and track sequences, N = 3 mice, *Figure 5—figure supplement 1J* and *Table 4*). These observations suggest that the primary role for the eye movements is stabilizing the visual fields.

If mice successfully track and capture prey by retaining the target in front of them then this should be reflected in the trajectories taken by the mice during the tracking epoch compared to the non-tracking behavioral epochs. During cricket tracking periods, mice ran directly toward the cricket, and their trajectories were significantly straighter than during equivalent non-tracking phases (*Figure 5C–G*). Lateral deviation at the half-way point (*Figure 5E*, non-tracking 4.3 ± 4.0 cm, tracking 1.4 ± 2.0 cm, p = 0.009), maximum lateral deviation (*Figure 5F*, non-tracking, 7.7 ± 4.9 cm, tracking 2.8 ± 2.0 cm, p = 0.0006) and the area between the trajectory and minimum distance path to the target (*Figure 5G*, area under the curve, non-tracking 135.6 ± 102.7 cm$^2$, tracking 51.3 ± 45.8 cm$^2$,

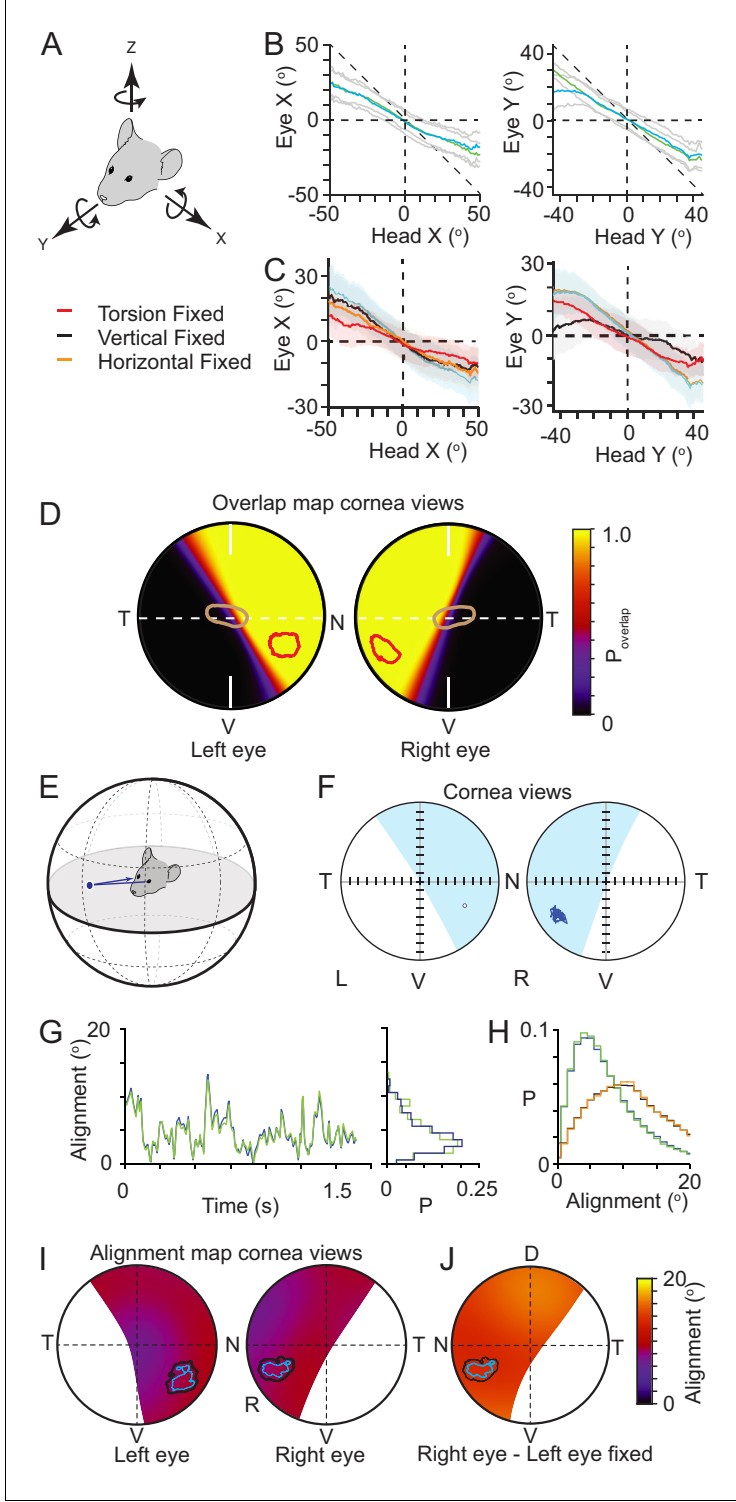

**Figure 4.** Functional foci are located within binocular regions in which motion is stabilized.  (**A**) Schematic of the common head and eye rotational axes. (**B**) Relationship between head and eye rotations around the common X (left, 154,625 frames from three animals) and Y (right, 165,432 frames from three animals) rotational axes during pursuit and non-pursuit sequences. Plots show mean for left (blue) and right (green) eyes with standard deviation (gray). (**C**) Relationship between head and left eye rotations around the common X (left) and Y (right) rotational axes with; all eye rotations present (blue), torsional eye rotations frozen (red), vertical eye rotations frozen (black) or horizontal eye rotations frozen (orange). Plots show mean (lines) and standard deviations (colored filled regions).
*Figure 4 continued on next page*

*Figure 4 continued*

(**D**) Corneal view showing probability of overlap of left and right visual fields for one example animal (71995 frames), with overlay of isodensity contours (red) from functional foci (see *Figure 2—figure supplement 1D*) and contour of second highest RGC region (brown) from *Figure 3B*. (**E**) Schematic of inter-ocular alignment. (**F**) Corneal view of alignment reference point in left eye (left) and variability in alignment of the re-projection of that point in the right eye (right) for a 1.6 s data segment. (**G**) Kinetics (left) and associated distribution (right) of the variability in ocular alignment for left eye point projected to right eye (blue) and right eye point in left eye (green) for one example data segment (shown in G) from one animal. (**H**) Distributions of ocular alignment from all data segments (159,318 frames, n=3 mice) with the measured eye movements for left into right eye (blue) and right into left eye (green) and alignment with eye movements frozen (left into right eye, black, right into left, orange). (**I**) Map of average inter-ocular alignment for all data segments (159,318 frames, n=3 animals) with overlay of isodensity contours from *Figure 2H*. (**J**) Map of average inter-ocular alignment as in J with left eye movements frozen.

The online version of this article includes the following source data and figure supplement(s) for figure 4:

**Source data 1.** Related to *Figure 4B,C*.
**Source data 2.** Related to *Figure 4D*.
**Source data 3.** Related to *Figure 4G*.
**Source data 4.** Related to *Figure 4H*.
**Source data 5.** Related to *Figure 4I*.
**Source data 6.** Related to *Figure 4J*.
**Figure supplement 1.** VOR relationships between head and eye rotations and abrupt shifts in gaze.
**Figure supplement 2.** Ocular torsion during cricket pursuit.
**Figure supplement 3.** VOR relationships between head and eye rotations and alignment of left and right eyes.

p = 0.0029) were all significantly smaller in the tracking epochs (all comparisons mean ± SD, N=13 tracking and non-tracking sequences, N=3 mice, Wilcoxon's Rank Sum Test).

Together this suggests that mice do not make compensatory vertical head movements, tracking eye movements or vergence eye movements to keep prey within their functional foci, but instead retain their target within a restricted bearing by running straight towards it. This raised the question of what is unique about the position of the functional focus on the cornea?

## Functional foci are located in region of minimized optic flow

Optic flow is the pattern of object motion across the retina that can be self-induced, through eye, head or translational motion, or induced by motion of objects in the environment, or combinations thereof (for review see: *Angelaki and Hess, 2005*). In a freely moving animal in a still environment,

**Table 3.** Compensation gain of eye rotations for head X or Y-axis rotations.
Effect of digitally freezing torsional, vertical, and horizontal eye rotations on the gain of compensation of X and Y head rotations. Data taken from 168,852 frames, from three animals.

| Eye | Rotation direction | Rotation | All Rotations (mean ± SD) | Eye rotation frozen (mean ± SD) |
|---|---|---|---|---|
| Left | X | Torsion | −0.45 ± 0.12 | −0.24 ± 0.1 |
| | | Horizontal | −0.45 ± 0.12 | −0.32 ± 0.06 |
| | | Vertical | −0.45 ± 0.12 | −0.35 ± 0.08 |
| Right | X | Torsion | −0.48 ± 0.06 | −0.24 ± 0.01 |
| | | Horizontal | −0.48 ± 0.06 | −0.36 ± 0.08 |
| | | Vertical | −0.48 ± 0.06 | −0.34 ± 0.03 |
| Left | Y | Torsion | −0.51 ± 0.12 | −0.35 ± 0.05 |
| | | Horizontal | −0.51 ± 0.12 | −0.51 ± 0.11 |
| | | Vertical | −0.51 ± 0.12 | −0.16 ± 0.14 |
| Right | Y | Torsion | −0.62 ± 0.05 | −0.45 ± 0.05 |
| | | Horizontal | −0.62 ± 0.05 | −0.62 ± 0.02 |
| | | Vertical | −0.62 ± 0.05 | −0.17 ± 0.03 |

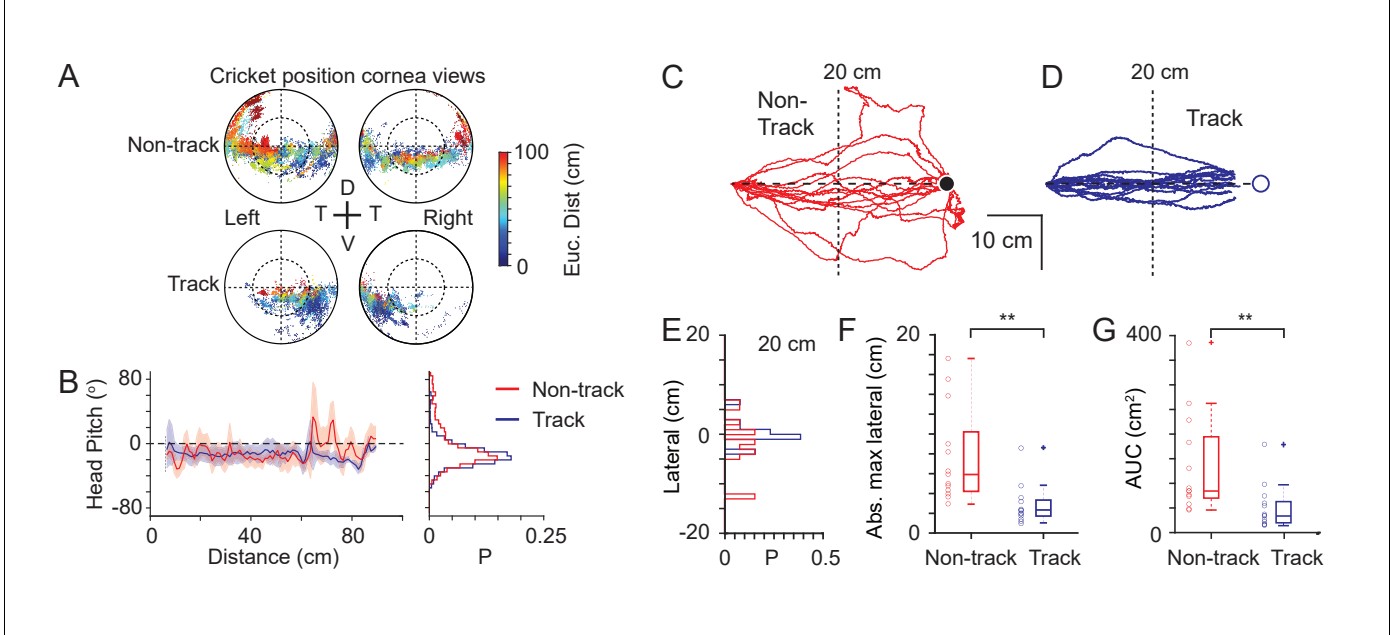

**Figure 5.** Mechanisms used to maintain prey within a focal visual region. (A) Corneal locations of cricket position color-coded by Euclidean distance to cricket for non-track (upper) and track (lower) epochs (18 data sequences, 15649 non-tracking and 8510 tracking frames, n=3 animals). (B) Mean and SD head pitch with Euclidean distance to cricket (left) and distribution of head pitch angles (right) for non-track (red) and track (blue) epochs (datasets as in A). (C) Mouse trajectories during non-track epochs rotated and overlaid to show deviation from a direct path (13 trajectories from three animals). (D) Mouse trajectories as in C but during track epochs (13 trajectories from three animals). (E) Histogram of lateral deviations for non-track (red) and track (blue) data in C and D calculated 20 cm from the end of the trajectory. (F) Boxplots and individual data points of absolute maximal lateral deviation from a direct path between start and end points for non-track (red) and track (blue) epochs (datasets as in C and D), ** p = 0.0006, Wilcoxon's Rank Sum Test. (G) Boxplots and individual data points of area under the curve (AUC) of mouse trajectories during non-track (red) and track (blue) epochs (datasets as in C and D), ** p = 0.0029, Wilcoxon's Rank Sum Test.

The online version of this article includes the following source data and figure supplement(s) for figure 5:

**Source data 1.** Related to *Figure 5A–G*.

**Figure supplement 1.** Eye movements during non-tracking and tracking periods.

**Table 4.** Eye rotations during non-tracking and tracking periods.
Horizontal, vertical, and torsional eye rotations during the non-tracking and tracking periods in *Figure 5*. Data taken from 18 non-track epochs and 18 track epochs, from three animals.

| Ocular Rotation | Non-Trk (mean ± SD) (median) | Track (mean ± SD) (median) | p value (KS) | P value (Student T-test) |
|---|---|---|---|---|
| Lt Horizontal | −1.8 ± 9.9° (−1.7°) | −1.8 ± 14.9° (−3.5°) | $3.9 \times 10^{-2}$ | 0.162 |
| Lt Vertical | 0.8 ± 11.2° (−0.4°) | 4.5 ± 11.1° (4.9°) | 0.425 | 0.616 |
| Lt Torsional | 2.9 ± 16.1° (0.0°) | 1.3 ± 20.6° (0.0°) | 0.945 | 0.610 |
| Rt Horizontal | 5.7 ± 10.9° (5.5°) | 1.0 ± 9.9° (1.7°) | $9.82 \times 10^{-2}$ | $1.08 \times 10^{-2}$ |
| Rt Vertical | −3.6 ± 13.4° (−6.3°) | 5.6 ± 12.7° (−7.1°) | 0.945 | 0.804 |
| Rt Torsional | 0.32 ± 13.5° (0.0°) | 0.7 ± 12.3° (0.0°) | 0.425 | 0.366 |

translational motion results in a pattern of optic flow that consists of a radial flow-fields emanating from a point of zero-optic flow (*Figure 6A*). While optic flow is used by many species for both navigation and the estimation of the motion properties of moving objects, motion induced blur degrades image formation on the retina and decreases resolution depending on the animal's direction of travel (*Land, 1999*). Optic flow is minimized in the direction of travel directly in front of the animal (*Sabbah et al., 2017*), with flow fields directed away from the travel direction and forming a second minimum directly behind the animal's head (*Figure 6A*, see also *Angelaki and Hess, 2005*). To measure the characteristics of optic flow in in both eyes of freely moving mice and to relate this flow pattern to the location of the functional foci, we next calculated average optic flow from freely moving data during pursuit behavior using the digitized environment and eye-views (*Figure 6B*). First, we calculated optic flow in the idealized case of forward translation motion when all surrounding surfaces were equidistant (*Figure 6C*). As mice have laterally facing eyes (optical axis = 59.9 ± 19.8° and −62.3 ± 14.7° lateral of frontal for the left and right eyes respectively, N=3 mice), idealized forward motion resulted in the region of minimal optic flow in each eye being located off optical axis in the ventro-medial corneal region representing the animal's forward direction (radial displacement from optical axis 36.64 ± 0.92° and −41.11 ± 2.27°, rotational angle 122.95 ± 17.05° and −107.94 ± 9.96°, for the left and right eyes respectively, mean ± SD, N=2 mice, *Figure 6C*). During free movement both the distance from the eyes to objects in the environment, as well as head and eye-rotations had a strong influence on the optic flow fields. We visualized the average flow fields during free motion by calculating the optic flow on the cornea during multiple pursuit trials (N=20 prey chases containing 52 tracking sequences, initial Euclidean distance mouse-cricket >20 cm). The resulting optic flow density maps were complex with a wide range of average speeds (133.44 ± 221.42 °/s, mean ± 1SD, median 28.64 °/s, interquartile range 4.57–137.18 °/s, N=2 mice, *Figure 6D*). The area of lowest optic flow extended from the nasal part of the field of view to overhead (*Figure 6D*) but, unlike the simulated case (*Figure 6C*), optic flow was not symmetric around the regions of minimal optic flow. Optic flow in the 30x30° region surrounding the ventro-medial point of minimal optic flow was significantly lower than that in an equivalent region in the ventro-temporal region during free movement, but not in the simulated case (free movement: nasal 46.3 ± 9.8 °/s, temporal: 199.4 ± 29.0 °/s, p = 0.0014, simulated: nasal 163.6 ± 82.2 °/s, temporal: 833.0 ± 416.5 °/s, p = 0.0662, mean ± SD, two-sided t-test, unequal variance, N=2 mice). Optic flow was higher in the lower visual field and considerably lower in the upper visual field (lower left eye visual field: 262.44 ± 106.50 °/s, upper left eye visual field: 44.87 ± 24.31 °/s, p = 1.78x10$^{-20}$, lower right eye visual field: 361.91 ± 168.80 °/s, upper right eye visual field: 40.59 ± 22.79 °/s, p = 6.68x10$^{-19}$, Two-sided t-test, unequal variance, N=2 mice), due to the greater distance between ceiling and mouse (distance to floor 2 ± 1 cm, distance to ceiling 308 ± 107 cm, 9873 frames, N=3 mice). Given the advantage of low optic flow to mammalian vision, we next quantified the position of least optic flow during prey tracking. We calculated the location of the translational optic flow minimum in each frame for each eye, and created a probability map of this location over the visual field (*Figure 6E*). The region of highest likelihood for the presence of the optic flow minimum overlapped considerably with the functional foci in both eyes during the tracking epochs of the pursuit behavior (overlap of optic flow 95% minima and functional foci 50% regions: 100% and 99 ± 1%, overlap of optic flow 50% minima and functional foci 50% regions: 61 ± 14% and 72 ± 4% in left and right eyes respectively, N=3 mice, *Figure 6E*). Together this shows that the behavioral strategy employed by mice during hunting, consisting of orienting themselves to directly face the prey and following a straight and direct course to it, results in the image of their prey coinciding with the region of reduced optic flow during pursuit, where the retinal image of their prey is least distorted due to motion induced image blur.

## Discussion

We developed a technique for reconstructing the visual fields in a freely moving mouse during prey pursuit to quantify the spatial relationship between the environment, cricket and the mouse. Using this approach, we show that during pursuit of crickets, the hunting behavior employed by mice results in the image of the prey consistently falling within a localized region of their visual field, termed here the functional focus. The positional maintenance of the cricket was not achieved by active eye movements that followed the prey, but rather by the animal's change in behavior, specifically the head-movement and orientation toward the prey during pursuit. While eye rotations

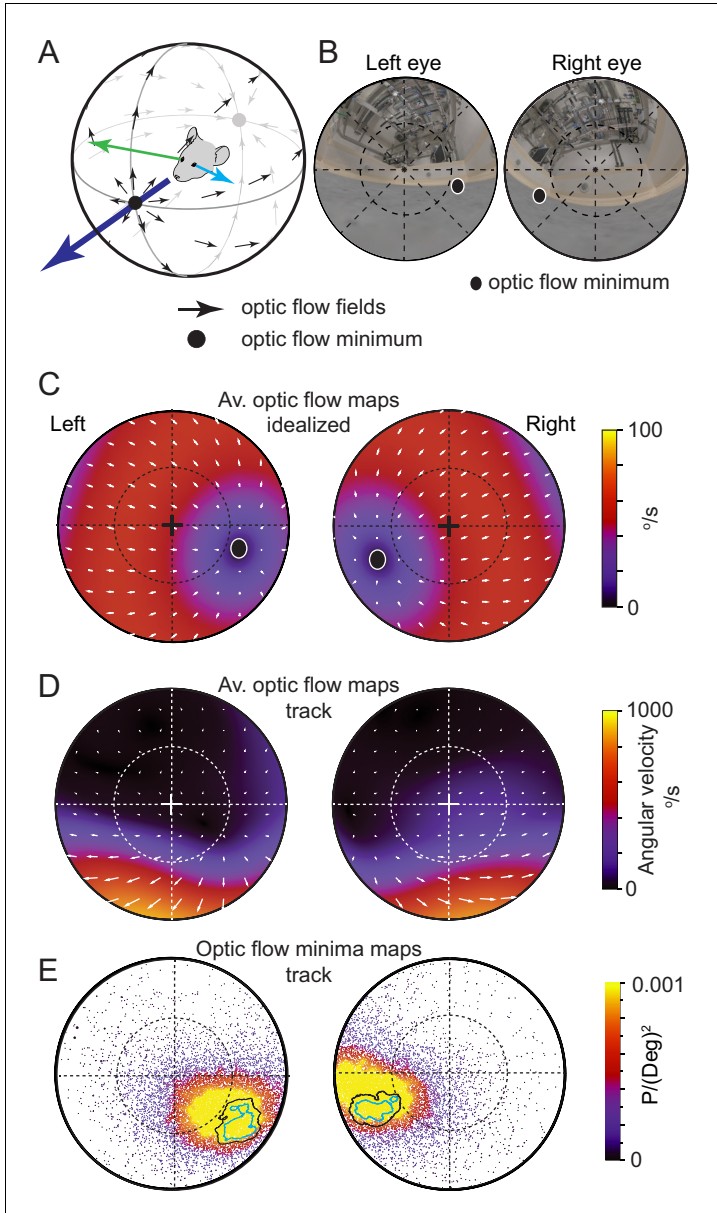

**Figure 6.** Functional foci are located in the regions of reduced optic flow during forward motion. (**A**) Schematic of idealized optic flow (black arrows) as a mouse translates forwards (after *Sabbah et al., 2017*). Left (blue arrow) and right (green arrow) gaze vectors. (**B**) Location of optic flow minima in reconstructed mouse eye views of the cricket and experiment arena (from *Figure 1H*), dashed circle represents 45°. (**C**) Optic flow map in corneal views, showing flow velocity (color coding) and direction (white arrows) calculated for the idealized spherical environment in 6A with forward motion of 50 cm/s. (**D**) Optic flow maps in corneal views during track epochs (5269 frames), from one animal. (**E**) Probability density map of optic flow poles in mouse corneal views during track epochs (data as in *Figures 2G*, 13,641 frames), with overlay of isodensity contours from *Figure 2H*.

The online version of this article includes the following source data for figure 6:

**Source data 1.** Related to *Figure 6C*.
**Source data 2.** Related to *Figure 6D*.
**Source data 3.** Related to *Figure 6E*.

stabilized the visual field via the vestibulo-ocular reflex by countering head rotations, the rotations were not specific to either prey detection or prey tracking. This strongly suggested that eye-rotations in mice, like in rats, primarily stabilize their large field of view and that all three rotational axes, including ocular torsion, combine to counter head rotations. In addition, we also show that eye rotations cannot be predicted from head rotations in any one axis as has been suggested by recent studies of mouse eye motion (*Meyer et al., 2020*; *Meyer et al., 2018*; *Michaiel et al., 2020*) but rather by a combination of all head rotations (*Figure 4—figure supplement 2*). As the eye rotations were predominately associated with countering head-rotations, this raised the question of whether the mouse can use a large fraction of its stabilized visual field to pursuit crickets, or whether a specific region is utilized. To accurately determine the correspondence between the animal's visual field and the retinal image, we developed a quantitative model of the mouse eye and optics. Using this we show that the location of the functional focus occurs within a dorso-temporal retinal region in an area with the highest density of Alpha-ON sustained RGCs, whose general properties have been previously proposed to be well suited for this purpose (*Bleckert et al., 2014*). Finally, we used the detailed, digitally rendered reconstruction of the arena and surrounding room to calculate the realistic optic flow in the visual fields (*Gibson et al., 1955*; *Sabbah et al., 2017*; *Saleem, 2020*) of the mice as they pursued crickets, which showed that the functional foci coincide with the region of the visual fields with minimal optic flow during the cricket pursuit, and presumably are thereby minimally distorted by motion-induced image blur (for review see *Angelaki and Hess, 2005*). Critical to this finding was the ability to isolate the visual sense, generate a detailed reconstruction of both the local environment and the animal's ocular anatomy and optical pathways, but also record eye motion in all three optical axes especially ocular torsion, something that has only been achieved in rats (*Wallace et al., 2013*). Lastly, by building an optical model and establishing the relationship between the retinal surface and the corneal surface we were able to relate the data generated from published studies on retinal anatomy (*Bleckert et al., 2014*; *Dräger and Olsen, 1981*; *Sterratt et al., 2013*) and physiology (*Dhande et al., 2015*; *Martersteck et al., 2017*; *Murphy and Rieke, 2006*; *Pang et al., 2003*; *Sabbah et al., 2017*; *van Wyk et al., 2009*) to our behavioral data.

Both estimates of the field of view of the mouse eye (*Dräger, 1978*) and electrophysiological measurements of receptive field locations of visually responsive neurons (*Dräger and Olsen, 1980*; *Wagor et al., 1980*) have established that the binocular region of the visual field in mice is contained within the nasal visual field of each eye, and spans a region of 30-40° in front of the animal (*Wagor et al., 1980*). We present here, that similar to the rat (*Wallace et al., 2013*), the overlapping monocular fields that make up the binocular overlap are not constantly maintained (*Figure 4H*) but fluctuate at the margins as the eyes rotate to counter head rotations (*Figure 4D*), resulting in a region where there is a transition from one area with near continuous binocular coverage, through to a region that is invariably monocular. The functional focus described here lies within the region of high probability of maintained binocular overlap. This region of the visual field projects onto the temporal retina, which contains both ipsilaterally projecting (uncrossed) RGCs (*Dräger and Olsen, 1980*; *Reese and Cowey, 1986*) and RGCs which form part of the callosal projection pathway (*Laing et al., 2015*; *Olavarria and Van Sluyters, 1983*; *Ramachandra et al., 2020*), both of which are considered central to binocular visual processing. In addition, the current study adds to the significance of these previous findings and suggests that the functional focus location is well placed to support stereoscopic depth perception, assuming that this form of visual processing is available to and employed by the mouse (*La Chioma et al., 2019*; *La Chioma et al., 2020*; *Samonds et al., 2019*; *Scholl et al., 2013*; *Scholl et al., 2015*). Further supportive of the importance and relevance of the region of binocular overlap, another recent study provides strong evidence to suggest that ipsilaterally projecting RGCs in the ventro-temporal retina are important in the final phase of cricket pursuit (mouse to cricket distance less than 6 cm), with selective ablation of these RGCs reducing the probability that coming into close proximity with the cricket resulted in its capture (*Johnson et al., 2021*). This finding is complimentary to the current study, in that it deals with the section of the hunting behavior excluded from analysis in the current study, that being behavioral segments where the distance between mouse and cricket is < 3 cm. This criteria was used in the current study to mitigate the possibility that the mouse was using its mystacial whiskers to detect or assist in detection of the cricket location. In the current study, we find that the location of the image of the cricket systematically shifts nasally and slightly ventrally on the cornea (temporally and slightly dorsally on the retina) as the mouse closes in on the cricket, and this may place the cricket's image

within the retinal region containing the ipsilaterally-projecting RGCs. As the mouse closes further, beyond our distance threshold, these RGCs may become increasingly important, particularly when the cricket is within grasping distance, where binocular vision and stereopsis may be most relevant.

While the overall highest density of retinal ganglion cells in mice is located in the region around the optical axis (*Dräger and Olsen, 1981*), a recent study examining the distributions of various different subclasses of RGCs has shown that the highest density of Alpha-ON sustained RGCs resides in the superior-temporal retina (*Bleckert et al., 2014*) in a region which would approximately coincide with the functional focus. These Alpha-ON sustained RGCs have center-surround receptive fields, a rapid response and fast conducting axon, and are thought to be 'spot detectors' (for review see *Dhande et al., 2015*). In addition, the Alpha-ON sustained RGCs in this particular retinal region differ from the same RGC-type in other regions of the retina as they have a significantly smaller dendritic tree radius and subtend a smaller area of physical space as well as have overlapping receptive fields (*Bleckert et al., 2014*). Taken together, the cellular properties as well as the region in-front of the animal which provides their input are consistent with the requirements for tracking small and mobile targets (*Bleckert et al., 2014*; *Dean et al., 1989*; *Lettvin et al., 1959*; *Procacci et al., 2020*). A recent study has shown that both wide-field and narrow-field neuronal types in the mouse superior colliculus play central roles in the detection and pursuit phases of this pursuit task, respectively (*Hoy et al., 2019*), and consistent with this, Alpha-ON sustained RGCs having projections to the superior colliculus (*Martersteck et al., 2017*). It is currently unclear how the primary visual cortex (V1) contributes to this behavior, but some role is possible if not probable, which would also be supported by the strong Alpha RGC projection to the dorsal lateral geniculate nucleus and thus V1 (*Martersteck et al., 2017*). Additionally, an increased cortical magnification factor occurs in the region corresponding to the nasal, binocular visual field (*Garrett et al., 2014*; *Schuett et al., 2002*).

Finally, we show that the region that contains these Alpha-ON sustained RGCs also coincides with the region of minimum optic flow and therefore reduced image blur during translation pursuit, a feature which would support accurate localization of small targets by Alpha-ON sustained RGCs. Patterns of optic flow are thought to be an important component of perception of self-motion (*Lappe et al., 1999*). Mechanistically supporting this, global alignment across the retina of the preferred orientation of direction-selective retinal ganglion cells with the cardinal directions of optic flow during idealized motion has been shown in mice (*Sabbah et al., 2017*). The average optic flow measured here was, perhaps not surprisingly, strikingly different from that observed with idealized motion, resulting in large part from the large differences to objects in the environment in which the behaviors were performed. For fast moving, ground dwelling animals like mice, considerable asymmetry in optic flow across the visual field may be the more normal case, considering that objects above the animal are, in general, likely to be more distant.

In freely moving rats, it has been shown that ocular torsion is correlated with head pitch such that nose-up rotation of the head is counteracted by incyclotorsion (rotation toward the nose) of both eyes, with nose-down pitch counteracted by excyclotorsion (*Wallace et al., 2013*). These rotations have the effect of stabilizing the horizontal plane of the retina with respect to the horizon. The considerable radial separation between the optical axis of the eye and both the functional foci observed in the current study as well as the highest density region of Alpha-ON sustained RGCs (*Bleckert et al., 2014*) renders the direction in which these regions point highly sensitive to torsional rotation. Consequently, torsional rotation also has an important effect on alignment of the left and right visual fields in addition to its role in visual field stabilization. We show here that torsional rotation in freely moving mice is also dynamic, with episodes showing in- and excyclovergence as well as dextro- and levocycloversion. Further, while the correlation between torsional rotation and head pitch observed in rats was measured, there was also an additional relation between ocular torsion and head roll consistent with VOR-evoked dextro- and levocycloversion. Consequently, prediction of torsion using a model based on head pitch alone resulted in an average error of around 7°, while an expanded model including roll as well performed better (*Figure 4—figure supplement 2J–O*).

In summary, we show here that during pursuit in mice the image of the intended prey falls consistently in a localized region of their visual fields, referred to here as the functional focus, and that this occurs through the animal orientating their head and body and running directly toward the prey rather than with specific eye movements. The location of the functional focus is within the binocular visual field, but in addition also coincides with the region of minimal optic flow during the pursuit, and presumably also minimally distorted by motion blur.

# Materials and methods

**Key resources table**

| Reagent type (species) or resource | Designation | Source or reference | Identifiers | Additional information |
|---|---|---|---|---|
| Software, algorithm | Matlab | Mathworks | Matlab 2019b | |
| Software, algorithm | OpenJDK | Oracle | Version 1.8.0_292 | |
| Software, algorithm | Cuda | Nvidia | Release 10.1, V10.1.243 | |
| Software, algorithm | Python | Python Software Foundation | Python 3.8.10 | |
| Software, algorithm | Qt | Qt Project | Qmake 3.1, Qt 5.9.5 | |
| Software, algorithm | OpenGL | Khronos Group/Nvidia/AMD | Version 4.6.0 | |

## Animal details

Experiments were carried out in accordance with protocols approved by the local animal welfare authorities (Landesamt für Natur Umwelt und Verbraucherschutz, Nordrhein-Westfalen, Germany, protocol number 84–02.04.2017.A260). Experiments were carried out using male C57Bl/6J mice acquired from Charles River Laboratories. At the time of the cricket hunting experiments, mice (n=9) were between 2 and 8 months old, and weighed between 21 and 29 g. Mice were maintained on a 12 hr light/dark cycle. Crickets (*Acheta domesticus*, Bugs-International, Germany) were housed in 480x375x210 cm cages with *ad lib* water and food (powdered mouse chow). During experiments in which head and eye rotations were recorded cricket body sizes ranged from 1 cm to 2 cm (1.8 ± 0.3 cm, mean ± SD, n=25).

## Implant surgery

Animals were anesthetized using fentanyl, medetomidine, and midazolam (50 µg/kg, 5 mg/kg, and 0.5 mg/kg, delivered i.p., respectively), and analgesia was provided with carprofen (7 mg/kg delivered s.c.). Body temperature was maintained using a thermostatically regulated heating pad. Respiration rate and depth of anesthesia was monitored throughout the procedure. Following opening of the skin and removal of connective tissue overlying the sagittal suture and parietal bones, the skull was cleaned with $H_2O_2$ (3%). A custom-made implant, consisting of a flat circular attachment surface for attachment to the skull, and implant body with three anti-rotation pins and a magnet (*Figure 7A–B*), was fixed to the dried skull using a UV-curing dental adhesive (Optibond FL, Kerr Corporation, Orange, California, USA) and a UV-curing dental composite (Charisma, Kulzer GmbH, Hanau, Germany). The implant attachment surface and body were made from light-weight, bio-compatible dental resin (Dental SG, Formlabs, Germany). Skin margins were closed with 5/0 Vicryl sutures (Ethicon Inc, Somerville, NJ, USA) and a cyanoacrylate adhesive (Histoacryl, B.Braun, Melsungen, Germany). The injectable anesthetic combination was antagonized with naloxone, atipamezole and flumazenil (respectively 1.2 mg/kg, 0.5 mg/kg and 0.75 mg/kg, delivered i.p.), and the animal was allowed to recover.

## Positioning of the head-mounted cameras

The eye cameras for oculo-videography were mounted on mounting arms which were attached to a baseplate with complementary holes to the anti-rotation pins on the implant and fitted with a magnet of complementary polarity. During positioning of the head-camera, mice were anesthetized with isoflurane (induction: 3–5% isoflurane, maintenance: 2.0% isoflurane in air). Anesthetic depth and body temperature were monitored as above. The cameras were positioned to have a sharp image of the entire eye, with the mounting arms adjusted such that the cameras and mounting system caused minimal disruption to the mouse's lateral and frontal field of view. Mounting arms were secured with cyanoacrylate adhesive glue (Histoacryl, B.Braun, Melsungen, Germany). The eye-camera system was then removed and the animal allowed to recover.

## Training procedure

Mice were acclimated to cricket capture in their home cage. Individual crickets were placed in the mouse's home cage overnight, in addition to their standard ad lib mouse food. Mice were handled

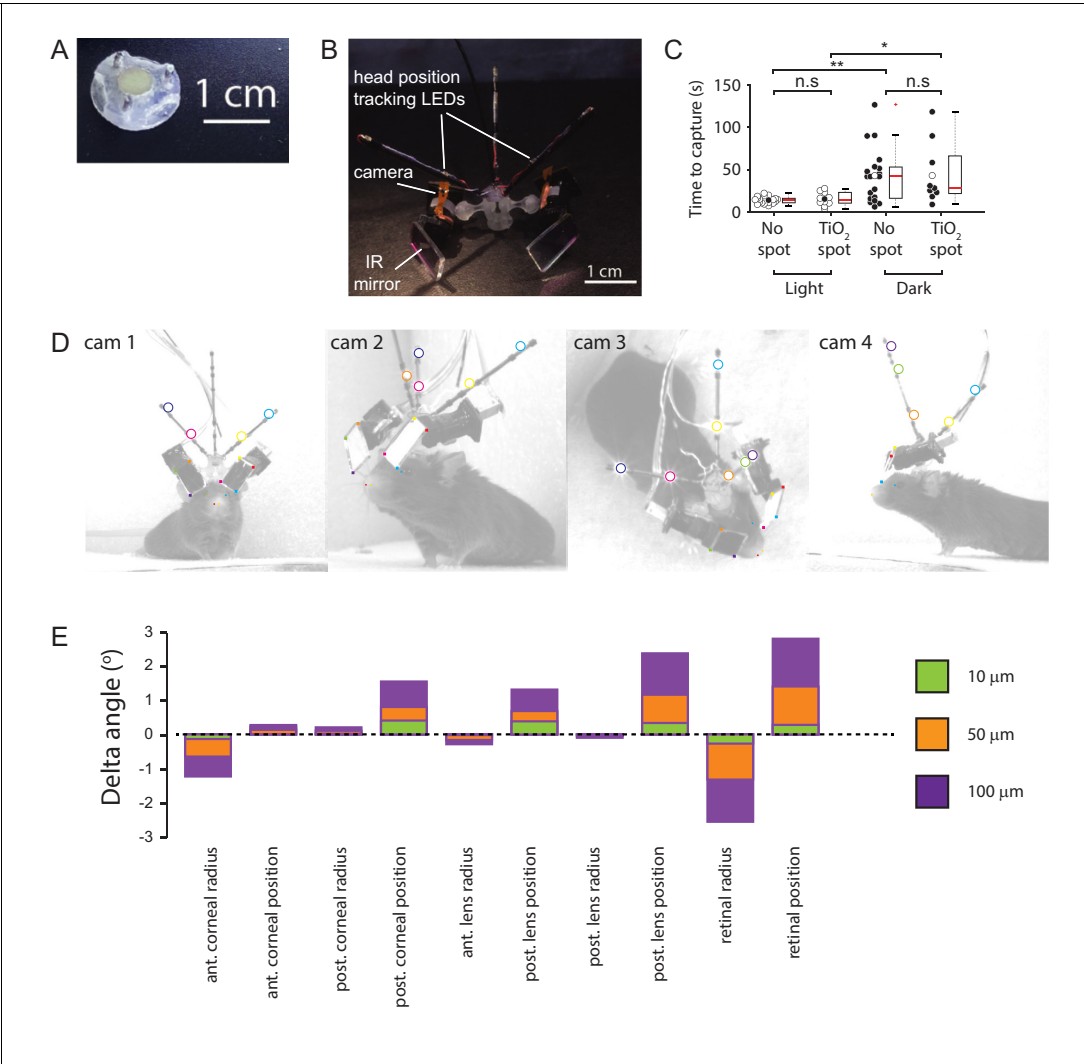

**Figure 7.** Methods. (**A**) Implanted baseplate with magnetic attachment point and restraining pin holes. (**B**) Miniaturized eye cameras and head position tracking system. Infrared illumination LEDs were mounted on the camera objective and reflected onto the eye using an IR-reflective mirror. Head position tracking IR-LEDs were mounted on three carbon-fiber struts attached to the head-mount. (**C**) Cricket capture times in lit or dark conditions in mice without (n=19 pursuit sequences, n=6 mice) or with (n=10 pursuit sequences in lit conditions and n=9 pursuit sequences in the dark, n=3 mice) corneal $TiO_2$ torsion tracking spots, Lit vs Dark with no spot, p = 0.0012, Lit vs Dark $TiO_2$ spot, p = 0.0133, Lit without spot vs Lit with $TiO_2$ spot, p = 0.69, Dark without spot vs Dark with $TiO_2$ spot, p = 1. n.s. = non-significant, *p<0.05, **p<0.01. Paired Wilcoxon's signed rank tests. For these experiments, pursuits were conducted in a smaller arena (480 x 375 x 210 cm). (**D**) Images of mouse with eye camera and head position tracking system for anatomical calibration. Head mount and anatomical features marked. Anatomical features: Left (blue filled circles) and right (green filled circle) medial canthi, left (orange filled circles) and right (red filled circle) nostril positions. Head mount features: position tracking LEDs (large colored circles), IR mirror corner positions (small colored filled squares). (**E**) Sensitivity of the radial elevation on the retina in the mouse eye model to changes in the radii of curvature and thicknesses of the model optical components.

The online version of this article includes the following source data for figure 7:

**Source data 1.** Related to *Figure 7C*.
**Source data 2.** Related to *Figure 7E*.

and habituated to the experimenter, the head cameras, and the head tracking mounts. The ability of each mouse to visually track the crickets was assessed using the protocol of *Hoy et al., 2016*. Briefly, the ability of the mice to track and capture crickets in a white walled, 480 x 375 x 210 cm arena was assessed in lit and dark conditions (*Figure 7C*). Mice were given 2 min in which to capture the crickets. Prior to the assessment mice were food deprived overnight before the trial.

## Placement of torsion tracking marks

Crenellations along the iridial-pupil border were less distinct in mice than those previous described in rats (*Wallace et al., 2013*). Ocular torsion changes were therefore measured by tracking the rotations of small spots of titanium dioxide ($TiO_2$) paste dots (~300 μm) applied to ventral and/or temporal locations on the cornea as described in *van Alphen et al., 2010*. The $TiO_2$ paste consisted of $TiO_2$ powder (Kronos Titan GmBH, Leverkusen, Germany) mixed with a small quantity of sterile artificial cerebrospinal fluid solution with the following composition (in mM): 135 NaCl, 5.4 KCl, 1.8 CaCl2, 1 MgCl2, 5 HEPES, pH balanced to 7.2 (300 mOsm/l). Application of the $TiO_2$ spots was performed with the animal anesthetized with isoflurane (induction: 5% isoflurane, maintenance: 0.5–1.0% isoflurane in air, total time anesthetized 5-10mins). Anesthetic depth and body temperature was monitored as above. Following application of $TiO_2$ spots, mice were allowed to recover for >45 min prior to a cricket hunt. The presence of the $TiO_2$ marks did not significantly change the animal's cricket hunting performance as assessed by the average time taken to capture crickets (*Figure 7C*).

## Experiment procedure

Initially, mice were allowed to explore the experimental arena (1x1x0.26 m) without head camera mounts. During subsequent training sessions, mice were acclimated to cricket hunting, with the head cameras on, in the experiment arena. Auditory white noise (60–65 dB, NCH-Tone generator v 3.26, NCH Software, Inc Greenwood Village, USA) was provided through four speakers (Visaton, Germany), one on each wall of the arena. Olfactory noise was provided by ventilating the arena (TD-1000/200 Silent fan, S and P, Barcelona, Spain) through a perforated floor (5 cm perforation spacing) with air blown through a cage containing live crickets (cricket cage dimensions 480x375x210cm). During experiments the arena was lit by a single lamp (4000 K, 9W, Osram, Munich, Germany) positioned ~1 m above the arena. During each experiment, the mouse was given 5 min to explore the arena without head cameras. After this period, the mouse was removed from the arena and the head cameras were mounted. At the commencement of each hunt, the cricket was released at a variable location into the central region of the arena.

## Eye camera and head position tracking system

Head and eye tracking was performed as described in *Wallace et al., 2013*, with modifications as described below. The eye camera mount and implant were re-designed to be smaller, lighter and stronger (*Figure 7A–B*). The camera system body, camera holders and mounting arms were produced using a Formlabs Form2 SLA 3D printer (Formlabs Inc, USA), with Dental SG Resin (Formlabs Inc, USA) as the primary construction material. The cable used for position tracking LEDs power inputs and for data transfer and camera were custom cables (Axon Kabel GmbH, Leonberg, Germany) combined with custom-designed flexible flat cables (IBR Ringler, Bad Rappenau, Germany) for the cameras, to reduce stiffness over the last 30 cm. Eye movements were recorded at 60 Hz (camera resolution 752x480 pixels), with illumination provided by a ring of three IR-LEDs (λ=850 nm, OSRAM SFH4050 or SFH4053 @ 70mA, RS Components, Germany) surrounding the camera lens. The mouse's head position and head rotations were tracked using seven IR-LEDs (λ = 950 nm, OSRAM SFH4043 @ 70mA, RS Components, Germany) mounted on three struts of carbon fiber that projected from the body of the camera system. The resultant total system weight was ~3 g, including effective cable weight.

## Mouse head and cricket position tracking

The positions of the cricket within the arena were recorded using four cameras (488 x 648 px, recorded at 200 Hz, piA640-210gm, Basler cameras, Basler Ahrensburg, Germany) fitted with NIR-blocking filters (Calflex X, Qioptiq, Germany). Cameras were located ~1.5 m above the arena and were positioned so that the arena was covered at all points by two or more cameras from differing vantage points. Mouse IR-head tracking LEDs were recorded at 200 Hz using four cameras (piA640-210gm, Basler cameras, Basler Ahrensburg, Germany). Image acquisition, synchronization, and mouse head rotation calculations were performed as described previously (*Wallace et al., 2013*).

## Anatomical model

Head mount features and mouse anatomical features (medial canthi and nostril positions) were recorded at 50 Hz using four synchronized cameras (acA2040-90um, Basler cameras, Basler Ahrensburg, Germany) fitted with 25 mm focal length objectives (CCTV lens, Kowa Optical Products Co. Ltd, Japan) calibrated as described below. Cameras were positioned to provide images of the animal and headset from different angles to allow triangulation of the anatomical features (*Figure 7D*). During acquisition of the calibration images, the animal was illuminated with 12 IR-LED modules, ($\lambda$ = 850 nm, Oslon Black PowerStar IR-LED module, ILH-IO01-85ML-SC201_WIR200, i-led.co.uk, Berkshire, UK) run at 1 A. Position tracking LEDs, medial canthi, nares, mirror corner and camera chip corner positions were marked in two or more camera views, in multiple synchronized frames. Based on the triangulated positions of anatomical features, head cameras and position tracking LEDs the mouse's eye position could be placed a common coordinate system.

To establish the animal's horizontal plane from the head tracking LEDs, a position for the animal's nose was first defined by averaging to 3D positions of the marked nostrils. A pre-forward vector was calculated using the direction between mean of eyes and nose and a pre-up vector as vector orthogonal to the pre-forward and vector between the eyes. Next, the left vector was defined as orthogonal to pre-forward and pre-up. Finally, the system was rotated by 40° around the left vector such that forward vector was elevated. This established a head-fixed forward-left-up coordinate system that was based on the bregma-lambda sagittal plane by tilting the eyes-nose plane by an angle of 40°.

## Interpolation

Head tracking frame rates were 200 Hz, while eye tracking cameras recorded at 60 Hz. Eye positions were consequently interpolated as follows: Let

$$R_{t_1}, R_{t_2} \in SO(3)$$

be two rotations that transform the vector $(0,0,-1)^t$ into the gaze vectors $v_{t1}$, $v_{t2}$ in head fixed coordinates at times $t_1$, $t_2$. Then for a time $t'$ with

$$t' = t_1 + s \cdot (t_2 - t_1), \quad 0<s<1$$

the corresponding rotation $R_{t'}$ is interpolated such that $v_{t'}$ is placed on the geodesic defined by $v_{t1}$, $v_{t2}$ with an angle of $s * \angle(v_{t1}, v_{t2})$ to $v_{t1}$, and the rotation of a vector perpendicular to $(0,0,-1)^t$ is continuous and uniform between $t_1$ and $t_2$.

## Camera calibration

Overhead cameras for animal position and cricket tracking, were calibrated as previously described for the overhead cameras (*Wallace et al., 2013*), with the addition of automated detection of corresponding points in the calibration images using openCV and the eye camera calibration performed as described in *Wallace et al., 2013*.

## Pupil position and pupil torsion tracking

Pupil boundary tracking, compensation for eye image displacement, and gaze vector calculation was performed as described previously in *Wallace et al., 2013*. Where contrast between pupil and iris was insufficient to allow automated pupil position tracking, pupil positions were manually tracked.

The TiO$_2$ spots for tracking ocular torsion were tracked manually in each image frame. Torsional rotations were determined based on the tracked TiO$_2$ spot positions as follows. Total rotation of the eye was defined as previously described in *Wallace et al., 2013*, as:

$$R_{\text{eye}} = R_\phi R_\theta R_\psi$$

$$= \begin{bmatrix} 1 & 0 & 0 \\ 0 & \cos\phi & -\sin\phi \\ 0 & \sin\phi & \cos\phi \end{bmatrix} \begin{bmatrix} \cos\theta & 0 & -\sin\theta \\ 0 & 1 & 0 \\ \sin\theta & 0 & \cos\theta \end{bmatrix} \begin{bmatrix} \cos\psi & -\sin\psi & 0 \\ \sin\psi & \cos\psi & 0 \\ 0 & 0 & 1 \end{bmatrix}$$

where $\phi$ = vertical, $\theta$ = horizontal and $\psi$ = torsional rotations. The mouse's gaze vector has the coordinates $\begin{bmatrix} 0 & 0 & -1 \end{bmatrix}^T$ for the reference position of the eye, and in each frame:

$$v^{\text{gaze}} = R_{\text{eye}} \begin{bmatrix} 0 \\ 0 \\ -1 \end{bmatrix}$$

With the eye in its reference position, we assume that the marked TiO$_2$ spot is located in the x-y plane of the eye camera (*Wallace et al., 2013*). The anatomical location of this marked spot can then be described by two unknown parameters $r$ (where r>1 is the 3D distance of the eyeball surface to the eyeball center, and a distance of 1 describes the rotation radius of the pupil) and $\alpha$ is the fixed angle between the TiO$_2$ mark and the gaze vector. After eye rotation the 3D location of the TiO$_2$ is:

$$v^{\text{mark}} = R_{\text{eye}} \begin{bmatrix} r\sin\alpha \\ 0 \\ -r\cos\alpha \end{bmatrix}$$

and the predicted pixel coordinates of the spot in the image are:

$$p^{\text{mark}} = \begin{bmatrix} a_{\text{EC}} \\ b_{\text{EC}} \end{bmatrix} + \frac{f}{z_0} \begin{bmatrix} v_1^{\text{mark}} \\ v_2^{\text{mark}} \end{bmatrix}$$

where $a_{\text{EC}}$ and $b_{\text{EC}}$ are the location in the image of the center of the eye ball and $\frac{f}{z_0}$ a scaling factor, both of which are determined in the calibration procedure for pupil boundary tracking, described in full in *Wallace et al., 2013*.

When $r$ and $\alpha$ are known the value $\psi$ can be determined. Using the Matlab function **fminbnd** the squared 2D distance

$$\left| p^{\text{mark}} \quad p^{\text{mark}} \right|_2^{c2}$$

between the predicted and marked locations of the TiO$_2$ mark is minimized.

This method is used to determine the ocular torsion based on the TiO$_2$ spot location, both during and after calibration. Calibration was performed as follows:

For a given $r$ and $\alpha$ choice, $\psi$ can be calculated as above. The sum of square errors in pixel locations is then calculated over all frames. We optimized over $r$ and $\alpha$ using the Matlab function **fminsearch**. To initialize $r$, we make use of the fact that the pupil model, $p^{\text{mark}}$ and $r$ together determine the 3D location of the mark $v^{\text{mark}}$ in each image. For each frame, we first calculated:

$$\Delta a = \frac{p_1^{\text{mark}} - a_{\text{EC}}}{f/z_0}$$

$$\Delta b = \frac{p_2^{\text{mark}} - b_{\text{EC}}}{f/z_0}$$

$$m = \min\left(1, \frac{r}{\sqrt{\Delta a^2 + \Delta b^2}}\right)$$

$$v_{\text{init}}^{\text{mark}} = \begin{bmatrix} m\Delta a \\ m\Delta b \\ -\sqrt{r^2 - m^2(\Delta a^2 + \Delta b^2)} \end{bmatrix}$$

$$\alpha_{\text{init}} = \cos^{-1}\left(v^{\text{gaze}}.v_{\text{init}}^{\text{mark}}/r\right)$$

Using this method, $\alpha_{\text{init}}$ is estimated separately for each frame, and if the choice of $r$ is correct then these values should agree. We can use **fminbind** to minimize the following with respect to $r$:

$$\mathrm{Var}(\alpha_{\mathrm{init}}) = \overline{(\alpha_{\mathrm{init}} - \overline{\alpha_{\mathrm{init}}})^2}$$

After $r$ is initialized, $\alpha_{\mathrm{init}}$ is calculated, with $\alpha$ initialized using the mean over frames.

Torsional rotations were normalized by calculating a mean torsion value for the 0.01% of frames that were closest to both median pitch and roll of the head. Torsional values in other tracked frames were then normalized to this mean torsion value.

## Cricket position tracking

Cricket body positions were automatically tracked using the method and algorithm described for tracking eye corners, as described in the section '*Compensation for lateral eyeball displacement – tracking of anatomical landmarks around the eye*' in *Wallace et al., 2013*. To increase the contrast between the region around the cricket in the image and the cricket, ~100 background image frames (in which neither mouse nor cricket was present) were averaged and subtracted from frames in which the cricket was present. In frames where automated cricket position tracking was not possible, frames were tracked manually. As the cameras used for cricket tracking had been calibrated along with the animal position tracking cameras (see above), the three-dimensional location of the cricket could be triangulated in a common coordinate system with the animal's position.

## Classification of behavioral periods

To decrease the effects of tracking noise and rapid head rotations, mouse velocity, target bearing and inter-animal Euclidean distances were first filtered using a 50 ms sliding window Gaussian filter.

The criteria used to classify the different hunt phases were based on those described in *Hoy et al., 2016*. In an initial step, behavioral end points ($t_{\mathrm{end}}$) for capture periods were identified by manual inspection of the tracking videos. Further identification of the behavioral start points ($t_{\mathrm{start}}$) and $t_{\mathrm{end}}$ points for the different hunt sequence epochs were then identified as described below.

The $t_{\mathrm{end}}$ points were defined as:

1. The $t_{\mathrm{end}}$ point for a detect period was defined as the last frame before (1) Mouse head velocity in the direction of the cricket was >= 20 cm/s, (2) The mouse's bearing towards the cricket was constantly below 90° and (3) the Euclidean distance between the mouse and cricket was continuously decreasing.
2. The $t_{\mathrm{end}}$ point for a tracking period was identified by locating local minima in the mouse-cricket Euclidean distance time plots, where local minima were defined as points at which the mouse came within a contact distance of 6 cm (measured from the tracked point on the mouse's head, giving a > 3 cm separation between the mouse's nose and the cricket). These were followed either by a capture period (see below) or were followed by a ⩾ 5 cm increase in inter-animal Euclidean distance, which were defined as cricket escapes. In cases where the absolute value of the target bearing was > 90° before the mouse turned towards the prey, the start of the tracking period was taken as the first frame in which the bearing to the target was <90°. Only tracking periods, in which the initial Euclidean distance between the mouse and cricket was >20 cm were analyzed.
3. The $t_{\mathrm{end}}$ point for the capture period was taken to be the point 6 cm away from the cricket, following which a cricket captured and consumed.

The start points of the hunt epochs were defined as follows:

1. The $t_{\mathrm{start}}$ for the detect period was the frame 500 ms prior to the detect $t_{\mathrm{end}}$ point.
2. The $t_{\mathrm{start}}$ for the tracking period was the first frame after the $t_{\mathrm{end}}$ detect frame.
3. The $t_{\mathrm{start}}$ for the capture period was either; (1) the first period in which the mouse approached the cricket and directly caught it, or (2) the first frame in which the mouse approached the cricket and all subsequent cricket escapes (prior to the final cricket capture) were less than 5 cm outside the contact zone (11 cm inter-animal Euclidean distance).

Cases in which the eye cameras were dislodged by the animal during the chase (n=4 hunt sequences) were included in the dataset up until the point where the cameras were dislodged.

### Target bearing

Target bearing was defined as the angle between the cricket position and the mouse's forward head direction in the horizontal plane.

### Digital reconstruction of arena

For the digital reconstruction, the company 3dScanlab (Cologne, Germany) was engaged to create a complete scan, photo series and 3D mesh model of the arena and room, which they performed using an RTC 360 3D laser scanner (Leica, Germany). The 3D point cloud produced by the laser scanner was converted to a 3D mesh model, to which textures of the experiment arena obtained from photographs (Nikon D810, 36 Mpx) were baked.

The camera tracking coordinate system, in which the mouse and cricket positions were tracked, and the scanned coordinate system of the 3D mesh model were aligned based on 16 fiducial points which could be clearly identified in both tracking camera images and the scan. Crickets were modeled as 2 cm diameter, 1 cm thick disks centered on their tracked position with the disk's axis oriented parallel to gravity.

### Generation of animal's eye view

Each eye was modeled as a hemisphere with a 180° field of view whose equator was perpendicular to the animal's gaze vector. For the projection of the environment onto the cornea, frame-wise animal's eye views for both eyes were created with custom written software in C++ (g++ 7.5.0, QMake 3.1, Qt 5.9.5, libopenexr 2.2.0, libpng 1.6.34 and OpenGL-core-profile 4.6.0) on a GeForce RTX 2070 (NVidia driver 450.66), using first cube mapping followed by a transformation into a spherical coordinate system. To do this, individual frame-wise coordinate transformations were made using the eye locations and orientations determined as described above to transform the mesh model of the arena and cricket to a static eye coordinate system using custom written vertex shaders to perform the coordinate transformation and the fragment shaders to texture the mesh. A cube-map (1024 x 1024 pixels per face) was created by performing such coordinate transformations for a 90 degree view in the direction of the optical axis of the eye and four mutually orthogonal directions. Custom written code was then used to transform the cube-map into a spherical coordinate system, with a 180 degree opening angle, using fragment shaders, resulting in a 1024 x 1024 pixel frame exported as png and OpenEXR files. In addition to the color map, maps of depth (pixel-wise object intersection distance), object identification, optic flow and 3D position of the object intersection point in the contralateral eye's coordinate system were also generated.

### Prey image probability density maps

For generation of the prey image probability density maps, animal's eye views were rendered that contained the cricket only (i.e. without inclusion of arena and room). Density maps from multiple detect-track sequences, and multiple animals, were made by averaging.

### Ocular alignment

Ocular alignment was defined as the consistency of the projection of a given point in the eye view of one eye into the other in an infinitely distant environment. This is equivalent to a projection in an idealized finite-distant spherical environment while assuming a distance between the animal's eyes of 0. For calculation, the radius of the sphere can then be set to 1 (without loss of generality). A point, located at the center of mass of the functional focus in each eye, was chosen from which to calculate the degree of inter-ocular alignment. This point was projected from one eye to the sphere surface and into the contralateral eye. The degree of alignment between the two eyes was calculated as follows:

Let

$$R_i, L_i : \mathbb{R}^3 \to \mathbb{R}^3$$

be the affine transformations for the left and right eye, and let

$$E \subseteq \mathbb{R}^3$$

be the idealized environment. For a given direction $u \in S^2$ we calculate the projection into the right eye $p_i \in R^3$ by:

$$p_i = L_i^{-1} \circ R_i \circ u$$

The average alignment is then calculated using the formula:

$$\overline{\sum} 2 \cdot \arcsin\left(\left|p_i - \left\langle\overline{\sum}p_i\right\rangle\right|/2\right)$$

where $\overline{\sum}p_i$ denotes mean and $\langle \rangle$ denotes normalization.

## Visual field overlap

Visual field overlap was analyzed in the idealized finite-distant spherical environment described above for ocular alignment. Visual overlap was calculated from the frame-wise maps of 3D object intersection points in the contralateral eye (see above section '*Generation of animal's eye view*') generated for the ocular alignment analysis: pixels whose 3D object intersection points had an angle of less than 90° to the optical axis were considered part of the overlapping field of view. Probability maps of overlap were calculated by averaging.

For analyses of the effect of freezing eye movements, eye rotations (horizontal, vertical, and torsional) were set to the mean rotation in one eye, and the effect quantified in the other eye view.

## Optic flow

To calculate the optic flow in a given pixel for a given eye, we consider the difference vector between the 3D positions in the static eye coordinate system of the object intersection point for this pixel one frame before and after the frame of interest, divided by $2 \cdot dt$ and mapped to unit distance by dividing by the distance between eye and interception point. This yields a 3D motion vector which is independent of influences of the frame rate and rendering resolution. The spherical projection used in the rendering process described above is a non-conformal, locally non-isometric map, meaning that angles between lines and distances between points are not preserved. This makes it necessary to evaluate the flow in each point in a local, orthonormal 3D coordinate system defined by the direction vector between the eye position and the object intersection point and derivative vectors along the angular coordinates $v\theta$ and $v\varphi$ at that point. Thus, we define the 2D flow at a given point as the orthogonal projection of the 3D flow vector onto the local plane spanned by $v\theta$ and $v\varphi$. In this study, we only use the first two components of the vector, while the third component contains the motion in radial direction to the eye.

In *Figure 5C*, optic flow was calculated for the animal in the idealized spherical environment described above, meaning the animal's head was equidistant to the surrounding at all points. This simplified scene was characterized as follows. Let

$$h \in \mathbb{R}^3$$

be the coordinate of the center of the mouse's head, then the scene around it was defined as

$$\left\{p \in \mathbb{R}^3 \,\middle|\, |p - h| = r\right\}$$

with r = 50 cm. For optic flow calculations, the sphere is considered fixed in global coordinates, and the flow is evaluated at the point where the mouse is in the center of the sphere translating forward at a speed of 1 cm/s.

In *Figure 5E*, optic flow was calculated with the animal in the digitally reconstructed environment (see above).

## Coloring of optic flow poles in mouse corneal views

The points in the scatter plot of optic flow poles in mouse corneal views were color-coded for the density of neighboring points using a two-dimensional Gaussian smoother with standard deviation

$$\sigma = \frac{2\pi}{180}$$

For a given point, the density was calculated as:

$$s_i = \sum_{j \in F} \frac{1}{2\pi\sigma^2} \exp\left(\frac{-|x_i - x_j|^2}{2 \cdot \sigma^2}\right) / |F|$$

where $F$ is the set of all considered frame indices, and

$$x_i = \frac{\partial_h[p]_i}{|\partial_h[p]_i|}$$

where $\partial_h[p]_i$ is the discrete central difference quotient of the mouse's eye trajectory $p$ in frame $i$, in the coordinate system of the respective eye, evaluated over h=4 frames.

## Mouse eye model

When constructing the eye model, we took experimentally determined values from *Barathi et al., 2008* (see *Table 1*). While we recognize that this study employed a different strain of mice to the one used here, the methodology used provides estimates of physical and optical parameters measured under conditions closest to those relevant for the current study. Further, variation of these parameters was not found to change the model to an extent that would influence the conclusions drawn from analyses involving the eye model (see below). These values distinctly define the spatial shapes and positions of the refractive components of the model eye (*Figure 3A*), as well as refractive indices for all but the lens, $n_{lens}$. We further assume a pupil radius of 594 µm, which is the mean of constricted and dilated mouse pupil sizes from *Pennesi et al., 1998*. We define the focal point of a bundle of rays as the point with minimal least squares distance to the rays. To optimize the missing refractive index $n_{lens}: \Omega \to \mathbb{R}^+$ inside the lens body $\Omega \subset \mathbb{R}^3$, we first calculated two lens models and optimized them such that the focal point of 10000 rays emitted from an object at 10 cm distance on the optical axis lay on the retina. The first model, for optimization of the lens surface, was derived with optimal constant refractive index $n_c \in \mathbb{R}^+$ over the volume. The second model, for lens gradient optimization, was derived with a smooth transition of refractive index to the anterior and posterior lens boundary, that is, $n_b = 1.333$ on $\partial\Omega$. We then used Poisson's equation $\delta n_g = c$, and optimized the strength of the gradient $c \in \mathbb{R}^+$. We assumed the final lens model as a linear combination of these two models:

$$n_{lens} = \alpha \cdot n_c + (1 - \alpha) \cdot n_g$$

with $\alpha \in [0, 1]$, where we optimized $\alpha$ as described for the above models, but from a point 10 cm away and 45° off optical axis. The derived refractive indices (*Table 2*) were within the range measured in *Cheng et al., 2019*.

To test the sensitivity of the model to changes in assumed physical parameters, we systematically changed the radius of curvatures listed in *Table 1*, and the thickness listed in *Table 2* by 10, 50, and 100 µm (several different values were used, to check the linearity of the dependence). We calculated the propagation of uncertainty through the eye model by analyzing the variation of radial elevation on the retina of the 45 rays (above), taking the numerical differentiation of each input variable that was used in the model. Lens optimization was performed for each newly generated eye model (as described above). The maximum deviations were 0.4, 1.38 and 2.76 degrees for the 10, 50, and 100 µm changes, respectively (*Figure 7E*), and overall, none of the observed effects on the model would influence the conclusions drawn from the analyses performed using the eye model.

## Projection from retina to cornea

The refractive elements in the rodent eye do not behave like ideally corrected optical elements, with the result that there is a distribution of incident rays with slightly varying angles of incidence on the cornea which converge on any given point on the retina. Projection from retina to cornea therefore requires an estimate of the distribution of outside world angles of incidence for any point of interest on the retina. To do this, we used a Monte-Carlo simulation to back-trace through the optics a set of

randomly chosen rays emerging from the point of interest on the retina. Since the intensity of light on a surface with an incoming angle of θ is proportional to $\cos(\theta)$, this function was also chosen for the probability density distribution of ray exit angles. The rays were then traced until they either hit any opaque surface, resulting in the affected ray being discarded, or passed through the anterior cornea, in which case the ray was accepted and its angle added to the distribution of passing exit angles for the respective point on the retina.

Refraction on boundary layers between different indices of refraction was performed analytically according to Snell's law. In volumes with a continuous variable refractive index (i.e. gradient-index (GRIN) optics), we used a finite-elements model. We first discretized the lens as a 40x40x40 lattice of side length 2.4 mm. We then started from initial conditions where s(0) is the point of incidence and v(0) is the vector of incidence multiplied by the speed of light c. The subsequent discrete trajectory and direction of propagation is then calculated step-wise according to

$$s(t_{i+1}) := s(t_i) + v(t_i) \cdot (t_{i+1} - t_i)$$

$$\tilde{v}(t_{i+1}) := \tilde{v}(t_i) + \nabla \log n(s(t_{i+1})) \cdot (t_{i+1} - t_i)$$

$$v(t_i) := \frac{\tilde{v}(t_i)}{|\tilde{v}(t_i)|^2}$$

The gradient is calculated in the lens lattice as the three-dimensional difference quotient, and then trilinearly interpolated to the exact position $s(t_i)$ of the ray.

## Projection of retinal ganglion cell density contours onto the model eye cornea

To determine the corneal location corresponding to the histologically identified retinal specialization in the mouse, isodensity lines were redrawn from *Dräger and Olsen, 1980* in Illustrator and digitized using Matlab. Isodensity lines enclosing regions containing the highest and second highest density of retinal ganglion cells, as well as the optic disc and outline of the retinal whole mount, were redrawn directly from Figure 3A in *Dräger and Olsen, 1981*, with horizontal being taken as horizontal (nasal-temporal) in the figure. The isodensity lines were scaled to match the eye diameter used for model eye, then placed into the model eye such that the center of mass of the optic disc reconstructed with the retinal ganglion cell contours was coincident with the intersection of the optic axis and retina in the eye model (*Figure 2—figure supplement 1A-C*). As the eye model was rotationally symmetrical, no further alignment between the histology and eye model was necessary. The high retinal ganglion cell density regions were then back-projected from retina to cornea as described above (*Figure 2—figure supplement 1D-E*).

## Eye in head coordinates

To quantify the effect of head rotations on VOR evoked eye movements in a common coordinate system, head rotations were normalized such that the average pitch and roll were 0. Axes were labeled X and Y respectively and eye rotations were represented using this horizon-aligned X-Y coordinate system. Positive head X values indicate head pitched up, while negative head X values indicate head pitched down. Negative head Y values indicate roll left, while positive Y values indicate roll right. Comparisons of the relationship between head and eye rotations were carried out using differential rotations between frame and average pose, defined in the following way:

$$\mathrm{l}' : \mathrm{L} \rightarrow \mathrm{G}, \mathrm{r}' : \mathrm{R} \rightarrow \mathrm{G}, \mathrm{h} : \mathrm{H} \rightarrow \mathrm{G}$$

are the affine transformations between Cartesian global coordinate system $\mathrm{G}$, head-fixed coordinate system $\mathrm{H}$ and left/right-eye coordiante systems $\mathrm{L/R}$.

The transformations from $\mathrm{L/R}$ respectively to $\mathrm{H}$ are:

$$\mathrm{l} = \mathrm{h}^{-1} \cdot \mathrm{l}'$$

$$r = h^{-1} \cdot r'$$

We calculate the left and right eye differential rotations as:

$$l_{\text{delta}} = l \cdot \bar{l}^{-1}$$

$$r_{\text{delta}} = r \cdot \bar{r}^{-1}$$

where $\bar{l}$ and $\bar{r}$ denote the average transformations over all frames (chordal L2 mean, implementation from SciPy 1.4.1).

## Statistical analysis

Within one experimental trial, the experimentally measured variables of interest are highly correlated with each other. This fact prevents us from using standard statistical tests on the whole time-trace to establish if any difference we observed in the data across different experimental conditions are significant or not, as one requirement of these kind on tests is that the samples from the populations being compared are independent of each other. However, we realized that trial-to-trial variability is the dominant source of variability in the data, whereas within-trial variability explains a smaller fraction of the total variance observed (a more detailed report is found in *Table 4*). For this reason, we decided to represent each temporal trace by its median value. We used the median and not the mean, because the former is more resistant to the presence of outliers and it is better suited to represent the 'average' value of a variable in this context. This operation reduced the size of the dataset to one data points per trial, which we can reasonably assume to be independent of each other.

## Acknowledgements

We thank Michael Bräuer, Rolf Honnef, Michael Straussfeld and Bernd Scheiding from the mechanical workshop for fabrication of the setup components. Arne Monsees and Florian Franzen for providing a version of the OpenCV based camera calibration software and Arne Monsees for helping with some aspects of the camera calibration. We also thank Samuel Akorli, Nada Eiadeh, Dennis Franzke, Julia Mauz, Kristina Mendeliene, Anastasia Nychyporchuk and Iftekha Hasan Siraje for assistance with pupil, head and cricket position tracking.

## Additional information

### Funding
No external funding was received for this work.

### Author contributions
Carl D Holmgren, Formal analysis, Validation, Investigation, Visualization, Methodology, Writing - original draft; Paul Stahr, Software, Formal analysis, Validation, Visualization, Methodology; Damian J Wallace, Software, Formal analysis, Supervision, Investigation, Visualization, Methodology, Writing - original draft, Writing - review and editing; Kay-Michael Voit, Software, Formal analysis, Supervision, Writing - original draft, Writing - review and editing; Emily J Matheson, Resources, Writing - review and editing; Juergen Sawinski, Resources, Investigation; Giacomo Bassetto, Formal analysis, Writing - review and editing; Jason ND Kerr, Conceptualization, Supervision, Funding acquisition, Visualization, Writing - original draft, Project administration, Writing - review and editing

### Author ORCIDs
Jason ND Kerr  https://orcid.org/0000-0001-9459-883X

### Ethics
Animal experimentation: Experiments were carried out in accordance with protocols approved by the local animal welfare authorities (Landesamt für Natur Umwelt und Verbraucherschutz, Nordrhein-

Westfalen, Germany, protocol number 84-02.04.2017.A260). Experiments were carried out using male C57Bl/6JCrl mice acquired from Charles River Laboratories.

## Decision letter and Author response
Decision letter https://doi.org/10.7554/eLife.70838.sa1
Author response https://doi.org/10.7554/eLife.70838.sa2

## Additional files

### Supplementary files
• Transparent reporting form

### Data availability

The mouse and cricket tracking data and resources for generating the reconstructed eye views are available on Dryad with DOI: https://doi.org/10.5061/dryad.2z34tmpnc.

The following dataset was generated:

| Author(s) | Year | Dataset title | Dataset URL | Database and Identifier |
|---|---|---|---|---|
| Holmgren C, Stahr P, Wallace D, Voit K, Matheson E, Sawinski J, Bassetto G, Kerr J | 2021 | Visual pursuit behavior in mice maintains the pursued prey on the retinal region with least optic flow | https://doi.org/10.5061/dryad.2z34tmpnc | Dryad Digital Repository, 10.5061/dryad.2z34tmpnc |

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
