## [Decision Letter]

**Acceptance summary:**

During natural behavior, like chasing prey, animals make continuous body, eye and head movements. These change the specific retinotopic regions that process the target stimulus. Holmgren et al., describe a technical tour de force that allows for reconstruction of what mice exactly see during natural behavior. They find that during prey hunting, the mice's movements are coordinated in such a way that mice keep the prey in a specialized retinotopic region. This region has characteristic properties that appear optimized for visual processing of the prey.

**Decision letter after peer review:**

Thank you for submitting your article "Freely-moving mice visually pursue prey using a retinal area with least optic flow" for consideration by *eLife*. Your article has been reviewed by 2 peer reviewers, including Martin Vinck as Reviewing Editor and Reviewer #1, and the evaluation has been overseen by Tirin Moore as the Senior Editor.

The two reviewers have discussed their reviews with one another, and the Reviewing Editor has drafted this to help you prepare a revised submission. Overall the reviewers were positive and appreciate the major technical tour de force to precisely monitor head and eye movements. The reviewers accept the statistical analyses and results. However they have concerns wrt the interpretation and to what extent the results are a consequence of the mouse simply running towards the cricket, or whether there is indeed something special about the specific binocular zone in which the cricket "lands". Addressing this should be a matter of enhancing the discussion, or if the authors want to make a strong claim adding some causal experiments.

Essential revisions:

1) A major concern is that readers will come away with the idea that the mouse is "foveating" the cricket and that there is something special about this location for optic flow, when both of these are consequences of the fact that the mouse is running toward the cricket. The authors should clarify what is active gaze control vs what is just a consequence of cricket pursuit. We encourage the authors that if they want to make a strong claim that this region is being actively used, there should be some causal support.

2) It would be useful to quantify whether the eye movements are strictly compensatory for head movements, or whether there are larger abrupts shift in gaze, as found in Michaiel.

3) Figures can be improved by labelling subpanels, as it is unclear what eg many of the circular plots represent without reading the caption.

4) The title is a bit misleading – it implies that mice are making active effort to use this region. Likewise, the abstract should note that both the specific retinal area and the optic flow pattern are direct consequences of running toward the cricket. And overall there is a lot of language that implies active targeting – the "the image of the prey is kept on a specific area". Since this is a consequence of the pursuit, rather than active maintaining the gaze, we suggest language more like "the image lands on a specific area" or "the image remains in a specific area".

5) If the authors want to support the inherent claim that this region is important for visually guided pursuit, which would provide more a significant advance, then some type of causal test would be helpful. Most conclusive would be a precise manipulation such as Johnson et al., 2020, e.g. manipulating α ganglion cells, or alternatively a lesion of the functional focus or blinders to occlude the functional focus.

6) The fact that the cricket does not land on the region of highest overall RGC density is a bit of an implausible strawman. Given that the cricket is in front of the mouse and that this region is located laterally (further than the amplitude of mouse eye movements) it's fairly clear that these will not overlap.

7) Johnson et al., (2021) provided a causal test demonstrating that mice not only use the binocular zone to pursue prey, but that the ipsi projecting RGCs that generate binocular receptive fields are necessary. This provides causal support for the findings demonstrated correlatively in this paper, yet this finding is not discussed beyond a reference to the fact that Johnson et al., also tracked head position during prey capture.

8) The authors emphasize that their setup eliminates auditory and olfactory cues. Is this essential to their results, i.e. do they find different targeting when these are not eliminated. Likewise, the title emphasizes visual pursuit, but does the cricket land on the same region when performing auditory pursuit (when controlled for pursuit accuracy?)

Reviewer #1 (Recommendations for the authors):

Holmgren et al., investigate how the head and eye movements in mice are coordinated during prey capture of crickets. A technical tour-de-force with eye and head tracking in freely moving animals allows for precise quantification of a cricket's position in the retinal field of view. Mice oriented towards the cricket using head rotations, maintaining the cricket in a narrow functional focus in the binocular, upper-temporal part of the retina. Eye movement were, as found in previous studies, primarily compensatory for head movements. The functional focus region has minimal optical flow and a high density region of α-ON retinal ganglion cells.

This is an impressive study, establishing a rich set of techniques for reconstruction of what the mice see during freely moving behavior, an approach that can be used for many different questions in the future. While some of the findings of the study are extensions of previous work, in particular, the compensatory head movements, the finding that during approach of the cricket it is kept within a relatively narrow functional focus with specialized properties is highly interesting. The finding will appeal to vision researchers, as well as researchers interested in the way in which our body optimizes sensory sampling of our environment according to ecological demands. The paper is well written and accessible to a broad audience.

An interesting question for future work would be to assess whether the head movements of the mice predict future trajectories of the cricket, thereby bringing the cricket into the predicted field of view, or whether the head movements are merely reactive to the cricket's behavior.

It would be useful to quantify whether the eye movements are strictly compensatory for head movements, or whether there are larger abrupts shift in gaze, as found in Michaiel.

A causal demonstration that silencing SC or V1 activity in this field of view disrupts behavior seems feasible and would provide direct support for the functional interpretations of the authors, but is not necessary for acceptance of the paper.

Figures can be improved by labelling subpanels, as it is unclear what eg many of the circular plots represent without reading the caption.

*Reviewer #2 (Recommendations for the authors):*

This study aims to determine whether mice use a specific region of their visual field, and consequently retina, to pursues a target during natural behavior, prey capture. This is an interesting question because mice do not have a clear fovea like primates, and it is significant in the field since the mouse Is a common model system for vision. Recent studies have shown that eye movements in the mouse are mostly compensatory with saccades serving to reset eye centering, and that eye movements and direction of gaze are primarily driven by head movements (Meyer et al., 2020), including in prey capture (Michael et al., 2020). Likewise, mice accurately aim towards prey during pursuit (Hoy et al., 2016), and thus the prey is in the binocular zone based on amplitude of eye movements (Michael et al., 2020). Furthermore, a recent study (Johnson et al., 2021) used ablation of ipsi-projecting retinal ganglion cells to show that the binocular visual field is necessary for prey capture.

The authors use a tour-de-force computational approach to determine the visual scene as projected on to the retina, based on 3-D scanning and virtual reconstruction of the environment, improved eye tracking methods, and computational projection of the scene onto the retina, all while mice pursue and capture crickets. This novel and rigorous approach, which will likely be useful to others in the field, is a significant advance and the real strength of the paper. However, as discussed below, most of the results they obtain from this effort do not provide a clear advance beyond what was known from previous studies, and are largely a direct consequence of mice running towards crickets during pursuit, though the impression one gets from the title and abstract is that this is an active and optimal gaze targeting strategy. One significant new piece of knowledge is that the cricket image lands on the region of highest density of α retinal ganglion cells, and this might be something to emphasize or explore more fully.

1. A main finding is that the cricket is generally maintained in a specific location in the lower binocular zone, which they term a "functional focus". Furthermore, the authors do several tests as to whether this is an active centering process and conclude that is not. "Mice do not make compensatory vertical head movements, tracking eye movements or vergence eye movements to keep prey within their functional foci but instead retain their target within a restricted bearing by running straight towards it". Thus, this main finding is just a direct consequence of the fact that mice are pursuing crickets by running towards them. And indeed two previous papers (Michael et al., 2020, Johnson et al., 2021) showed that mice keep prey in their binocular zone and maintain their head tilted downward, so this is not necessarily new information.

2. The authors also ask "what advantage is this behavior to the mouse?" The obvious answer is that running straight toward crickets is how to catch them. In other words, this is not a gaze targeting strategy but simply a consequence of that fact that if you want to catch something, you run towards it. However, instead the authors analyze the pattern of optic flow and demonstrate that this places the cricket in the region of least optic flow. Again, this is likely a trivial consequence of the fact that mice are running straight towards the cricket. Translational motion results in a focus of expansion (FOE) of optic flow in the heading direction, and the FOE itself is a minimum in the optic flow. Hence, as long as the mice are running toward the cricket, it will be in a region that is on average (given distortions of optic flow resulting from eye movements) a minimum of optic flow. Therefore I don't see this as a significant finding, and particularly as the title, unless they can demonstrate that this is not trivially true.

3. The title is a bit misleading – it implies that mice are making active effort to use this region. Likewise, the abstract should note that both the specific retinal area and the optic flow pattern are direct consequences of running toward the cricket. And overall there is a lot of language that implies active targeting – the "the image of the prey is kept on a specific area". Since this is a consequence of the pursuit, rather than active maintaining the gaze, I'd suggest language more like "the image lands on a specific area" or "the image remains in a specific area".

4. If the authors want to support the inherent claim that this region is important for visually guided pursuit, which would provide more a significant advance, then some type of causal test would be helpful. Most conclusive would be a precise manipulation such as Johnson et al., 2020, e.g. manipulating α ganglion cells, or alternatively a lesion of the functional focus or blinders to occlude the functional focus.

5. The fact that the cricket does not land on the region of highest overall RGC density is a bit of an implausible strawman. Given that the cricket is in front of the mouse and that this region is located laterally (further than the amplitude of mouse eye movements) it's fairly clear that these will not overlap.

6. Johnson et al., (2021) provided a causal test demonstrating that mice not only use the binocular zone to pursue prey, but that the ipsi projecting RGCs that generate binocular receptive fields are necessary. This provides causal support for the findings demonstrated correlatively in this paper, yet this finding is not discussed beyond a reference to the fact that Johnson et al., also tracked head position during prey capture.

7. The authors emphasize that their setup eliminates auditory and olfactory cues. Is this essential to their results, i.e. do they find different targeting when these are not eliminated. Likewise, the title emphasizes visual pursuit, but does the cricket land on the same region when performing auditory pursuit (when controlled for pursuit accuracy?)

[Editors' note: further revisions were suggested prior to acceptance, as described below.]

Thank you for resubmitting your work entitled "Visual pursuit behavior in mice maintains the pursued prey on the retinal region with least optic flow" for further consideration by *eLife*. Your revised article has been evaluated by Tirin Moore (Senior Editor) and a Reviewing Editor.

The manuscript has been improved but there are some remaining issues that need to be addressed before final acceptance, as outlined below. These can be addressed with textual changes. Please clearly indicate those changes so we can proceed with acceptance as soon as possible.

*Reviewer #1 (Recommendations for the authors):*

The authors have addressed all of my comments.

*Reviewer #2 (Recommendations for the authors):*

The authors have overall done an excellent job of revising their claims to match the data and avoid misleading conclusions by readers. I have two remaining points that weren't fully addressed.

1. The authors have generally improved the wording in the text to avoid implying that mice are actively targeting the cricket to this region. However, this is still not very clearly stated in the abstract, which is what most people take away from a paper. Indeed, in their response, the authors state that their findings are best stated in the first paragraph of the discussion as "The positional maintenance of the cricket was not achieved by active eye movements that followed the prey, but rather by the animal's change in behavior, specifically the head-movement and orientation towards the prey during pursuit." A statement along these lines in the abstract would be very effective, so that readers don't have to wait until the discussion to have this stated explicitly. If word length is the issue, I suggest finding something else to trim since as the authors themselves note, this is the best statement of the conclusion from their findings.

2. In the public section of the original review, I noted that the fact that the cricket lands in the region of least motion blur was an almost trivial consequence of the fact that the mouse is running towards the cricket, and hence the cricket is at the focus of expansion (minimum of optic flow). However, this has not been addressed in the revision – I apologize if that is because it was in the public comments rather than author comments. However, it seems essential to address this so that readers do not think that this is an "design principle" of mouse vision. Unless the authors can provide a compelling argument for how this could not be the case, given the geometry of optic flow, then the statement that it lands on the region of least motion blur should be accompanied, in the abstract and elsewhere, by the clarification that this is a direct consequence of motion towards the cricket rather than a specific optimality. I also suggest revising the title to emphasize a more compelling finding (e.g. α RGCs), but if the authors want to continue to use this title, accompanied by appropriate explanation in the abstract, then that is their prerogative.

---

## [Author Response]

Essential revisions:1) A major concern is that readers will come away with the idea that the mouse is "foveating" the cricket and that there is something special about this location for optic flow, when both of these are consequences of the fact that the mouse is running toward the cricket. The authors should clarify what is active gaze control vs what is just a consequence of cricket pursuit. We encourage the authors that if they want to make a strong claim that this region is being actively used, there should be some causal support.

We thank the reviewers for raising this point, because this is a central and important conclusion of the study. To state it clearly here, our conclusion is that mice do not “foveate” or have pursuit-like eye movements for following the cricket. We found no evidence to support the presence of pursuit-like movements, neither vergence movements (Figure 5 —figure supplement 1J) nor combinations of movements that would fulfill the role of keeping the cricket image in a consistent location on the cornea (Figure 5a) during the tracking phase of the behavior. Our conclusion was probably best stated in the first paragraph of the Discussion of the initially submitted manuscript as:

“…The positional maintenance of the cricket was not achieved by active eye movements that followed the prey, but rather by the animal’s change in behavior, specifically the head-movement and orientation towards the prey during pursuit. While eye rotations stabilized the visual field via the vestibulo-ocular reflex by countering head rotations, the rotations were not specific to either prey detection or prey tracking. This strongly suggested that eye-rotations in mice, like in rats, primarily stabilize their large field of view and that all three rotational axes, including ocular torsion, combine to counter head rotations. …”.

We agree with the reviewers that there were several sections of the submitted text that have the potential to give a reader an incorrect impression of our conclusion in this regard, and we have modified the text with the aim of making this point clearer and more consistent throughout the document. The text modifications are outlined below.

Abstract:

“…the behavior. By quantifying the spatial location of objects in the visual scene and their motion throughout the behavior, we show that the prey image consistently falls within a small area of the VOR-stabilized visual field. This functional focus coincides with the region of minimal optic flow in the visual field and consequently minimal motion-induced image blur during pursuit. The functional focus lies in the upper-temporal part of the retina and coincides with the reported high density-region of Α-ON sustained retinal ganglion cells.”

Introduction:

page 5: “…the binocular field and undertake direct pursuit. Prey objects remain in the functional foci through the stabilizing action of the VOR, and not through active prey-pursuit eye movements. The stabilized…”

Results:

page 7, heading: “During pursuit the image of the prey consistently falls in a localized visual region”

page 9: “…Mean Absolute Difference with bootstrapping, N=57 detect-track sequences, N = 3 mice. Thus, during the tracking and pursuit behavior the image of the prey consistently fell on a local and specific retinal area that we refer to from here on as the functional focus. The functional focus fell within the binocular field, while the region …”

page 18: “… N=3 mice, Figure 6E. Together this shows that the behavioral strategy employed by mice during hunting, consisting of orienting themselves to directly face the prey and following a straight, direct course to it, results in the image of their prey coinciding with the region of reduced optic flow during pursuit, where the retinal image of their prey is least distorted due to motion induced image blur.”

page 19: “…environment, cricket and the mouse. Using this approach, we show that during pursuit of crickets the hunting behavior employed by mice results in the image of the prey consistently falling within a localized region of their visual field, termed here the functional focus. …”

page 23: “In summary, we show here that during pursuit in mice the image of the intended prey falls consistently in a localized region of their visual fields, referred to here as the functional focus, and that this occurs through the animal orientating their head and body and running directly towards…”

2) It would be useful to quantify whether the eye movements are strictly compensatory for head movements, or whether there are larger abrupts shift in gaze, as found in Michaiel.

We did observe the abrupt shifts in gaze reported by Michiael et al., and by Meyer et al., 2020 and these are included in the trajectories and calculations. However, as these eye movements occur in conjunction with the animal’s yaw turns they neither formed a large fraction of total duration of the data we analyzed nor were they of direct consequence to the location of the cricket in the eye view during the tracking phase of the behavior, which was the primary interest in this manuscript. In addition, as these eye movements had been described previously in the above mentioned publications we elected not to isolate and analyze them. However, in response to this reviewer comment we have added a reference to these movements in the text and an example trace in figure 4 supplement 1. The text now reads as follows:

page 11: “…and counter head rotations through the VOR, enabling the large field of view around the animals head to be stabilized while the animal is moving. The relationships between head rotations and both the horizontal and vertical eye rotations have recently been quantified, and in addition it has been reported that both during exploration and hunting, mice also have abrupt gaze shifts brought about by the combination of head rotation and conjugate saccade-like horizontal eye rotations(Meyer, O'Keefe et al., 2020, Michaiel, Abe et al., 2020). We also observed both forms of eye movements in the current study (Figure 4 —figure supplement 1). However, how these rotations combine with torsional rotations is not known. If mouse …”

3) Figures can be improved by labelling subpanels, as it is unclear what eg many of the circular plots represent without reading the caption.

With the aim of making the figures clearer we have now added labels on figure panels 1IandJ, 2E-H, 3BandE, 4D, F and I, 5A and 6C-E, as well as figure 1 supplement 1 panel H, all panels in figure 2 supplement 1 and figure 4 supplement 1 panels H, J, KandL. We feel this is a reasonable balance between adding clarity to the figures while not overcrowding.

4) The title is a bit misleading – it implies that mice are making active effort to use this region. Likewise, the abstract should note that both the specific retinal area and the optic flow pattern are direct consequences of running toward the cricket. And overall there is a lot of language that implies active targeting – the "the image of the prey is kept on a specific area". Since this is a consequence of the pursuit, rather than active maintaining the gaze, we suggest language more like "the image lands on a specific area" or "the image remains in a specific area".

We appreciate the reviewers concerns regarding implications about active targeting, and have made a number of modifications to address this throughout the document.

First, we have modified the title to “Visual pursuit behavior in mice maintains the pursued prey on the retinal region with least optic flow”.

With respect to the abstract, we would like to point out that the localization of the functional focus is not only a consequence of the mouse running directly toward the cricket, but rather a consequence of this behavior and the visual-stabilizing effect of the vestibular-ocular reflex. We have modified the abstract with this point in mind, as described in our response to point 1. Further, we have also made numerous modifications throughout the text to address this concern as also described in our response to point 1.

5) If the authors want to support the inherent claim that this region is important for visually guided pursuit, which would provide more a significant advance, then some type of causal test would be helpful. Most conclusive would be a precise manipulation such as Johnson et al., 2020, e.g. manipulating α ganglion cells, or alternatively a lesion of the functional focus or blinders to occlude the functional focus.

While we agree with the reviewers that a causal demonstration would reinforce the conclusion of the study, we do not think that there is an experimental design which allows the causality to be unequivocally established in this case, and further, the additional time delay imposed by the need to obtain both approval for new experiments and to acquire and control the required mouse lines or other experimental devices would preclude timely publication of the results in this study.

In addition, the primary difficulty in performing an experiment designed to demonstrate the causal link between a retinal region or ganglion cell type or subtype and a behavioral outcome, is the requirement that it can be unequivocally shown that any behavioral impairment is really due specifically to the intended manipulation and not to simply the loss of vision in part of the visual field. To take the example of the study by Johnson et al., 2021, while their manipulation should indeed cleanly eliminate the function of the ventro-temporal ipsilaterally projecting RGCs, the wide distribution of these neurons across the retina shown in their figure 3 would indicate that this inhibition may cause a very broad loss or impairment of vision across the animals visual field. This visual loss alone may disturb the animals’ behavior rather than the behavioral impairment being due to a deficit specifically in binocular vision or stereopsis. That this hunting behavior is predominantly a visual behavior would seem to be clear, and consequently showing that the behavior was impaired by a manipulation causing some form of loss of vision is perhaps not particularly compelling. The same logic applies also to manipulations involving either blinkering of the visual field or a lesion. A manipulation that selectively targeted the Α-ON sustained subset of Α RGCs (not all Α RCGs have an enhanced density in the dorso-temporal region (Bleckert 2014)) would provide considerable strength to the conclusion, however, we are not currently aware of a mouse line that would allow this specific manipulation and even if it were available the administration associated with performing the experiment would prevent timely publication of the current result. We will of course try to follow up on this in future studies.

6) The fact that the cricket does not land on the region of highest overall RGC density is a bit of an implausible strawman. Given that the cricket is in front of the mouse and that this region is located laterally (further than the amplitude of mouse eye movements) it's fairly clear that these will not overlap.

We acknowledge the reviewers point, but maintain that the comparison of the locations of the functional focus and density contours for all ganglion cells is a reasonable and impartial comparison which is of general interest. The abundance of RGCs contributing to the density distribution for all RGCs is numerically two orders of magnitude greater, and presumably has a substantial biological significance, for what, is now not clear.

We have modified the introductory heading to this part of the Results section and substantially re-written it with this criticism in mind, and have removed the section dealing with the rotations and errors required for the functional focus and overall RGC density contours to overlap, as we agree with the reviewers that this comparison is not necessary. We have also modified a section of the discussion dealing with this topic, and removed 4 panels from figure 3 supplement 1 (panels H-K) which also dealt with this.

This section now contains numerous small and some larger modifications and has not been reproduced here for the sake of space, however the section begins on page 9 with the modified heading “Relative locations of functional foci and ganglion cell density distributions”.

The modified Discussion reads:

page 19: “…of the mouse eye and optics. Using this we show that the location of the functional focus occurs within a dorso-temporal retinal region in an area with the highest density of Α-ON sustained RGCs, whose general properties have been previously proposed to be well suited for this purpose (Bleckert, Schwartz et al., 2014). Finally,…”

7) Johnson et al., (2021) provided a causal test demonstrating that mice not only use the binocular zone to pursue prey, but that the ipsi projecting RGCs that generate binocular receptive fields are necessary. This provides causal support for the findings demonstrated correlatively in this paper, yet this finding is not discussed beyond a reference to the fact that Johnson et al., also tracked head position during prey capture.

We understand the point of the reviewers but we somewhat disagree with the strength of the conclusions possible from the Johnson et al., (2021) manuscript. Firstly, ablation of the ipsilaterally-projecting RGCs did not extinguish the behavior, but rather reduced the probability of a successful capture when the mouse was in close proximity with the cricket and extended the total time required for capture. They provide strong evidence that these RGCs are involved in close-range interaction between the mouse and cricket, and it may be the case that these RGCs are required for binocular viewing, binocular correspondence or stereopsis at this range. By the same argumentation, we further think that to call this a “causal” demonstration is perhaps extending the reasonable conclusions from the Johnson et al., study a little beyond their intention; the mice could still successfully pursue and capture crickets in the absence of functional ipsilaterally-projecting RGCs and it was not determined by the Johnson et al., study whether this was residual visual ability or alternatively guided by tactile or olfactory cues.

With regard to methodology, Johnson et al., do not measure eye position directly but infer the animals’ visual fields from its head orientation, which they measure from lateral imaging. Without high resolution imaging and 3D triangulation it is difficult to accurately measure 3D changes in head orientation, and with this in mind we note that their estimate of average pitch in their study is close to double that reported in the current study and by other studies (Oommen and Stahl 2008, Michaiel, Abe et al., 2020).

We do acknowledge that the Johnson et al., study is very complimentary to the current study, in that they focus on the final phase of the capture behavior (mouse-cricket distance < 6cm) while we specifically exclude this data (mouse-cricket distance < 3cm) to avoid any misinterpretation arising from the animal using its tactile sense through the mystacial whiskers. Further, the data presented in our Figure 5A suggests that as the mouse approaches the cricket, the cricket image systematically drifts further and further toward the dorsal tip of the retinal region in which the ipsilaterally-projecting RGCs reside, which is supportive of a potential role for these RGCs in the final phase of the hunt where the cricket is in close proximity to the mouse.

We have now included an extended description of the points outlined in the paragraph above in the Discussion section as follows:

page 21: “…and employed by the mouse (Scholl, Burge et al., 2013, Scholl, Pattadkal et al., 2015, La Chioma, Bonhoeffer et al., 2019, Samonds, Choi et al., 2019, La Chioma, Bonhoeffer et al., 2020). Further supportive of the importance and relevance of the region of binocular overlap, another recent study provides strong evidence to suggest that ipsilaterally-projecting RGCs in the ventro-temporal retina are important in the final phase of cricket pursuit (mouse to cricket distance less than 6 cm), with selective ablation of these RGCs reducing the probability that coming into close proximity with the cricket resulted in its capture (Johnson, Fitzpatrick et al., 2021). This finding is complimentary to the current study, in that it deals with the section of the hunting behavior excluded from analysis in the current study, that being behavioral segments where the distance between mouse and cricket is < 3cm. This criteria was used in the current study to mitigate the possibility that the mouse was using its mystacial whiskers to detect or assist in detection of the cricket location. In the current study we find that the location of the image of the cricket systematically shifts nasally and slightly ventrally on the cornea (temporally and slightly dorsally on the retina) as the mouse closes in on the cricket, and this may place the cricket’s image within the retinal region containing the ipsilaterally-projecting RGCs. As the mouse closes further, beyond our distance threshold, these RGCs may become increasingly important, particularly when the cricket is within grasping distance, where binocular vision and stereopsis may be most relevant.”

8) The authors emphasize that their setup eliminates auditory and olfactory cues. Is this essential to their results, i.e. do they find different targeting when these are not eliminated. Likewise, the title emphasizes visual pursuit, but does the cricket land on the same region when performing auditory pursuit (when controlled for pursuit accuracy?)

We conclude here that the functional focus observed is a consequence of both the behavioral strategy employed by the mice during pursuit and the VOR-based stabilization of the visual field during the animal’s movement. As mice use auditory, visual and olfactory cues during prey pursuit and can swap between modalities (Langley 1983, *Animal Behaviour, 31*, 199-205; Langley 1988, *Behav Processes, 16*, 67-73; Gire, Kapoor et al., 2016, *Curr Biol, 26*, 1261-1273) we removed any potential confound by conducting the experiments in the presence of olfactory and auditory noise. We did measure the time to capture with and without olfactory cues from the same mice and found that, as expected from the literature, without olfactory cues the mice took significantly longer than with olfactory cues present (Results section, page 6). There were no overt differences in behavior, but we did not record eye movements during these control experiments. Unless the animals changed their behavioral strategy when they have useful information from all senses available to them, we cannot see how the targeting would change, given the strong influence of the VOR on their eye position.

Regarding the image localization on the cornea in the case of auditory pursuit, while this is a very interesting point it is a bit beyond the scope of the current study. A reasonable initial expectation would be that the animal should employ a behavioral strategy for hunting which minimizes the energy expenditure during the pursuit and maximizes the probability of success. Assuming that the mouse can localize the cricket accurately, running directly towards it, as we observed in the current study, is presumably one way in which the required energy expenditure can be reduced or minimized. If we further assume that the mouse can localize the cricket accurately with its auditory sense alone, then the direct path to the cricket would be similar, resulting in similar targeting of the cricket in the eye view. This has the caveat that if the best orientation of the head for auditory-based localization is different to that for ideal visual localization, then the location of the cricket image on the cornea may be different as a consequence of the altered behavior. [Editors' note: further revisions were suggested prior to acceptance, as described below.]

Reviewer #2 (Recommendations for the authors):The authors have overall done an excellent job of revising their claims to match the data and avoid misleading conclusions by readers. I have two remaining points that weren't fully addressed.1. The authors have generally improved the wording in the text to avoid implying that mice are actively targeting the cricket to this region. However, this is still not very clearly stated in the abstract, which is what most people take away from a paper. Indeed, in their response, the authors state that their findings are best stated in the first paragraph of the discussion as "The positional maintenance of the cricket was not achieved by active eye movements that followed the prey, but rather by the animal's change in behavior, specifically the head-movement and orientation towards the prey during pursuit." A statement along these lines in the abstract would be very effective, so that readers don't have to wait until the discussion to have this stated explicitly. If word length is the issue, I suggest finding something else to trim since as the authors themselves note, this is the best statement of the conclusion from their findings.

Thank you for the additional comments for clarification. We have now changed the abstract to include the following sentence:

“This functional focus coincides with the region of minimal optic flow within the visual field and consequently area of minimal motion-induced image-blur, as during pursuit mice ran directly toward the prey.”

2. In the public section of the original review, I noted that the fact that the cricket lands in the region of least motion blur was an almost trivial consequence of the fact that the mouse is running towards the cricket, and hence the cricket is at the focus of expansion (minimum of optic flow). However, this has not been addressed in the revision – I apologize if that is because it was in the public comments rather than author comments. However, it seems essential to address this so that readers do not think that this is an "design principle" of mouse vision. Unless the authors can provide a compelling argument for how this could not be the case, given the geometry of optic flow, then the statement that it lands on the region of least motion blur should be accompanied, in the abstract and elsewhere, by the clarification that this is a direct consequence of motion towards the cricket rather than a specific optimality.

We thank the reviewers for raising this point, which we had unfortunately overlooked and not directly responded to in our previous author comments. We had/have amended sections of the text to clarify for the reader that the overlap of the prey image and the region of minimal optic flow arises as a result of the animal’s motion towards the prey. In particular:

1. We have changed the wording of the abstract (please see point 1).

We have added an ending to the following sentence:

2. Introduction, Line (83): “The stabilized functional foci are spatially distinct from the regions of highest total retinal ganglion cell density, which are directed laterally, but coincides with the regions of the visual field where there is minimal optic flow and therefore minimal motion-induced image disturbance during the behavior as the mouse runs towards the cricket.”

In addition, in light of the previous reviewer comments, we had modified the text to:

3. Results: “Together this suggests that mice do not make compensatory vertical head movements, tracking eye movements or vergence eye movements to keep prey within their functional foci but instead retain their target within a restricted bearing by running straight towards it.

4. Results: “Together this shows that the behavioral strategy employed by mice during hunting, consisting of orienting themselves to directly face the prey and following a straight and direct course to it, results in the image of their prey coinciding with the region of reduced optic flow during pursuit, where the retinal image of their prey is least distorted due to motion induced image blur.”

5. Discussion: “The positional maintenance of the cricket was not achieved by active eye movements that followed the prey, but rather by the animal’s change in behavior, specifically the head-movement and orientation towards the prey during pursuit.”

6. Discussion: “Finally, we show that the region that contains these Α-ON sustained RGCs also coincides with the region of minimum optic flow and therefore reduced image blur during translation pursuit, a feature which would supports accurate localization of small targets by Α-ON sustained RGCs.”

7. Discussion: “In summary, we show here that during pursuit in mice the image of the preferentially keep their intended prey falls consistently in a localized region of their visual fields, referred to here as the functional focus, and that this occurs through the animal but do so by orientating their head and body and running directly towards the prey rather than with specific eye movements.”

As to the question as to whether or not this is a “design principle” of mouse vision we would suggest that it is hard to differentiate what is by design and what is not. In general, there is a tradeoff between panoramic field of view and eye-position in the head, which in addition, at least for the mouse and rat, is stabilized by a strong VOR. These factors undoubtably aid the rodents in survival and in many respects dictate how vision is used by the animal. Together with the topographical differences in the retinal densities of some functionally important retinal cell types, this means that cells tuned for tuned some specific tasks will sample different regions of visual space (Baden et al., 2016; Bleckert et al., 2014; Hughes, 1977; Sabbah et al., 2017; Szatko et al., 2020; Franke et al., 2017; Zhang et al., 2012, Szel et al., 1992). Dorso-ventral differences in cone subtype aid contrast detection of objects in the sky (Baden 2013), W3 cells in the ventral retina aid detection of prey in the sky (Zhang et al., 2012) and the orientation and location of direction selective RGCs in the retina align with the orientation of idealized optic flow fields (Sabbah et al., 2017). Spectral differences in UV in ground vs sky and specific features of the aerial predator may be selective evolutionary pressures in the first two cases, while in the latter case, the characteristics of optic flow during forward translational motion may be a part of the evolutionary pressure influencing the direction selective RGCs. Similarly, for visualization of objects in the ventro-medial visual field, the properties of the optic flow fields produced as the mouse runs forwards towards its target may be a selective pressure on VOR-stabilized RGCs in the dorso-temporal retina.

With regards to the point that it seems ‘trivial’ that the location of the functional focus and the region of least optic flow coincide, our suggestion is that this results from a behavioral strategy employed by the mouse, that being to run directly at the cricket. This need not have been the case. The mouse could theoretically instead have used a strategy of predicting the future position of the cricket based on the cricket’s current trajectory and speed and run at that. This would result in the cricket being outside the region of minimal optic flow.

In both cases these points are more speculative and we have therefore not included them in the Discussion section.

I also suggest revising the title to emphasize a more compelling finding (e.g. α RGCs), but if the authors want to continue to use this title, accompanied by appropriate explanation in the abstract, then that is their prerogative.

While we are very curious about the overlap of the functional focus and the reported high density-region of Α-ONsustained retinal ganglion cells, we think that it is a bit premature at this stage to put this in the title. Consequently, we have elected to keep the current title but have changed the abstract as requested (please see point 1).